# Stabilizing Sample Similarity in Representation via Mitigating Random Consistency

Jieting Wang [1]    Zelong Zhang [1]    Feijiang Li [1]    Yuhua Qian [* 1]    Xinyan Liang [1]

## Abstract

Deep learning excels at capturing complex data representations, yet quantifying the discriminative quality of these representations remains challenging. While unsupervised metrics often assess pairwise sample similarity, classification tasks fundamentally require class-level discrimination. To bridge this gap, we propose a novel loss function that evaluates representation discriminability via the Euclidean distance between the learned similarity matrix and the true class adjacency matrix. We identify random consistency—an inherent bias in Euclidean distance metrics—as a key obstacle to reliable evaluation, affecting both fairness and discrimination. To address this, we derive the expected Euclidean distance under uniformly distributed label permutations and introduce its closed-form solution, the Pure Square Euclidean Distance (PSED), which provably eliminates random consistency. Theoretically, we demonstrate that PSED satisfies heterogeneity and unbiasedness guarantees, and establish its generalization bound via the exponential Orlicz norm, confirming its statistical learnability. Empirically, our method surpasses conventional loss functions across multiple benchmarks, achieving significant improvements in accuracy, $F_1$ score, and class-structure differentiation. (Code is published in https://github.com/FeijiangLi/ICML2025-PSED)

## 1. Introduction

The representation power (Goodfellow et al., 2016; Ghande-harioun et al., 2024) of deep learning refers to its ability to automatically learn and extract features from data through multi-layer neural networks, eliminating the need for manually designed features. Owing to this powerful capability, deep learning has been widely applied across various fields, including machine vision (Chen et al., 2020; Kondratyuk et al., 2024; Dosovitskiy et al., 2021), time-series signal analysis (Xu et al., 2022; Bian et al., 2024; Crabbé et al., 2024), and many others.

The strategies for enhancing network representation ability can be grouped into four categories. First, structural optimization improves feature extraction through multi-scale learning (He et al., 2016; Lin et al., 2017) or self-attention mechanisms (Vaswani et al., 2017). Second, data enhancement boosts model robustness through data augmentation and generative modeling, improving generalization (Goodfellow et al., 2014). Third, training process optimization prevents overfitting and enhances stability via multi-task learning and regularization (Srivastava et al., 2014; Ioffe & Szegedy, 2015; Ng, 2004). Finally, loss function optimization aims to guide effective learning by designing suitable loss functions (Rangapuram et al., 2018). Intuitively, evaluating the quality of sample similarity at the representation layer in a loss function can be an effective approach.

Recently, an unsupervised measure, $d_{infor}(\mathbf{K})$, was proposed to quantify the informativeness of similarity matrices (Brockmeier et al., 2017). It computes the distance between a similarity matrix $\mathbf{K}$ and a set of non-informative matrices. Let $\mathcal{N}_a$ be the set of non informative matrices $\mathcal{N}_a = \{(1-a)\boldsymbol{I} + a\boldsymbol{J}, 0 \leq a \leq 1\}$, $\boldsymbol{I}$ is the identity matrix, $\boldsymbol{J}$ is the full one matrix. The similarity matrix described by $\mathcal{N}_a$ represent scenarios where different samples exhibit uniform similarity. The measure $d_{infor}(\mathbf{K})$ is defined as:

$$d_{infor}(\mathbf{K}) = \min_{0 \leq a \leq 1} \|\mathbf{K} - \mathcal{N}_a\|_F^2, \qquad (1)$$

$$= \frac{1}{n^2}\|\mathbf{K}\|_F^2 - \frac{1}{n-1}(n\overline{\mathbf{K}}^2 - 2\overline{\mathbf{K}} + 1),$$

where $\overline{\mathbf{K}} = \frac{1}{n^2}\mathbf{1}^{\mathsf{T}}\mathbf{K}\mathbf{1}$, $n$ is the number of sample, and $\| \cdot \|_F^2$ is the Frobenius norm (the square root of the sum of

[1]Institute of Big Data Science and Industry, Key Laboratory of Evolutionary Science Intelligence of Shanxi Province, Shanxi University, Taiyuan, China. Correspondence to: Yuhua Qian <jinchengqyh@126.com>.

*Proceedings of the 42nd International Conference on Machine Learning*, Vancouver, Canada. PMLR 267, 2025. Copyright 2025 by the author(s).

squared elements of the matrix). This measure effectively captures the information embedded in $\mathbf{K}$ and can guide $\mathbf{K}$ to assign different similarity values to each pair of samples. However, the underlying assumption in classification tasks is that samples from the same class should exhibit higher similarity compared to those from different classes. The pairwise-based evaluation $d_{infor}(\mathbf{K})$ overlooks the broader class-level distinctions necessary for effective classification.

To evaluate the discriminative ability of the classification model, an intuitive measure and a novel loss function is the Square Euclidean Distance (SED), which compares $\mathbf{K}$ to the true adjacency matrix $\mathbf{Y}\mathbf{Y}^{\mathsf{T}}$,

$$d_{SED}(\mathbf{K}) = \|\mathbf{K} - \mathbf{Y}\mathbf{Y}^T\|_F^2, \tag{2}$$

where $\mathbf{Y}$ is the one-hot encoding of the true label $Y$, $\mathbf{Y} \in \{0,1\}^{n \times k}$ is the one-hot encoding of the true label vector, $k$ is the number of classes. However, SED is biased toward certain non-informative matrices, restricting its capacity to establish meaningful similarity relationships.

Consistency metrics measure the agreement between two random variables, while random consistency(RC) refers to spurious agreement arising purely from randomness (Wang et al., 2023). A canonical manifestation of RC occurs when examinees achieve measurable test scores solely via random response patterns. The mechanisms by which RC harms the learning process include evaluation distortion, optimization misguidance, and generalization barriers. Failure to deduct the RC baseline may lead to overestimating the model's actual consistency performance (e.g., an original consistency score of 0.6 vs. a random baseline of 0.2 means the true effective consistency should be 0.4). When loss functions include RC without proper correction, they can induce optimization bias, causing algorithms to spuriously improve consistency metrics by overfitting to noise (Wang et al., 2020a) or data bias (Li et al., 2024; Vinh et al., 2010) instead of learning genuine data patterns. These would consequently impair the model's generalization ability. The Pure Consistency Measure (PCM) framework (Wang et al., 2020a;b) addresses RC in metrics like accuracy (Wang et al., 2023) and the Gini index (Wang et al., 2024), mitigating decision and multi-value bias. In clustering, mitigating RC reduces cluster number bias (Vinh et al., 2010), and in causal learning, PHSIC (Li et al., 2024) reduces bias related to dimensionality and sample size.

To address RC in SED, we propose a novel Pure Square Euclidean Distance (PSED) under the Pure Consistency Measure framework. PSED refines SED by incorporating the expected distances of adjacency matrices generated through label permutations as a baseline. This measure can address the shortcomings of $d_{infor}(\mathbf{K})$ and $d_{SED}(\mathbf{K})$.

Theoretical analysis of our approach highlights two main advantages: improved heterogeneity and unbiasedness in

similarity matrix selection, ensuring more reliable representations of hidden layers. Furthermore, we provide a learning bound for PSED based on a statistical norm, offering theoretical guarantees on the method's generalization performance. In summary, the main contributions are as follows:

- A loss function for measuring the ability of the representation layer is proposed, and an explicit solution for the loss function in the version of eliminating random consistency is given.

- Through theoretical analysis, the advantages of this metric in heterogeneity and unbiasedness have been demonstrated, and a generalization bound has been provided for the generalization performance of the loss function in fully connected layer network structures.

- A fully connected network classification model based on this loss function was proposed, and the effectiveness of the algorithm was verified through extensive experiments.

The proofs and some experiment results are in Appendix.

## 2. Related Work

The main contents involved are loss function and generalization bound, and we will review these two aspects.

### 2.1. Loss Function in Deep Learning

Loss functions in deep learning measure the discrepancy between model predictions and actual values, guiding the optimization of model parameters. Common loss functions include Mean Squared Error (MSE) for regression tasks (LeCun et al., 2015), Cross-Entropy for classification (Hinton et al., 2012), Hinge loss for binary classification with SVMs (Cortes & Vapnik, 1995), Huber loss combining MSE and absolute error for robust regression (Huber, 1964), Kullback-Leibler Divergence for comparing probability distributions in generative models (Kingma & Welling, 2014), and Contrastive loss for evaluating sample similarity in metric learning tasks like face verification (Chopra et al., 2005). Selecting the appropriate loss function is crucial, and custom ones may be necessary for specific tasks. In this paper, we propose a metric to measure the quality of similarity matrices as a loss function to guide deep learning.

### 2.2. Learn ability

The generalization error represents the gap between the training error and the test error, with this bound capturing the factors influencing the test error. Existing traditional theories based on VC dimension and Rademacher complexity are insufficient to explain the performance of

deep learning (Vapnik & Chervonenkis, 1971; Bartlett & Mendelson, 2002). While numerous norm-based bounds have been proposed (Neyshabur et al., 2015; 2018; Bartlett et al., 2017; Golowich et al., 2018; Arora et al., 2018), we choose an exponential Orlicz norm-based concentration inequality (Vershynin, 2018). This choice is motivated by the fact that this norm characterizes the concentration behavior of the network parameters, rather than merely the range of parameter values considered by traditional norms. Furthermore, exponential Orlicz norm-based inequalities encompass traditional norm-based inequalities, as for variables with bounded values, their exponential Orlicz norms must also be bounded.

## 3. Definition and Analytic Solution

Given a hypothesis function space $\mathcal{F}$, the task of classification is to learn a function $h(X) \in \mathcal{F}$ that maps from the feature space $X \in \mathcal{X} \subseteq \mathbb{R}^d$ to the discrete label space $Y \in \mathcal{Y}$. To measure the representation ability of the function, we seek the Euclidean distance between the adjacency matrix of the true labels and the similarity matrix calculated by $h(X)$. We provide an analytical solution as follows:

$$d_{SED}(\mathbf{K}, \mathbf{Y}) = \|\mathbf{K} - \mathbf{Y}\mathbf{Y}^T\|_F^2 \qquad (3)$$
$$= \|\mathbf{K}\|_F^2 + \sum_{i=1}^{k} m_i^2 - 2 \sum_{i=1}^{k} \mathbf{1}_{m_i}^T \mathbf{K}_{[i][i]} \mathbf{1}_{m_i}$$

where $\mathbf{1}_{m_i}$ is single column all 1 vectors of length $m_i$, $m_i$ is the number of objects of $i$ class and $\mathbf{K}_{[i][i]}$ is the sub kernel matrix of objects in class $i$.

Since $d_{SED}$ is a consistency measure, previous work (Wang et al., 2020a;b) has shown that random consistency exists in consistency measures. To mitigate random consistency in Formula 28, we adopt the pure consistency framework.

For two random variables $Z_1, Z_2$, the framework of pure consistency measure (PCM) refers to eliminate random consistency from consistency measure (Wang et al., 2020a;b):

$$PCM(Z_1, Z_2) = CM(Z_1, Z_2) - RCM(Z_1, Z_2), \quad (4)$$

where $CM(Z_1, Z_2)$ represents the degree of consistency between random variables $Z_1$ and $Z_2$ and $RCM(Z_1, Z_2)$ represents the degree of consistency generated by chance.

Then we provide the definition of Pure Square Euclidean Distance (PSED) in the framework of random consistency:

**Definition 3.1.** The PSED is defined as:

$$d_{PSED}(\mathbf{K}) = d_{SED}(\mathbf{K}, \mathbf{Y}) - \mathbb{E}_{\mathbf{Y}'}(d_{SED}(\mathbf{K}, \mathbf{Y}')) \quad (5)$$
$$= \|\mathbf{K} - \mathbf{Y}\mathbf{Y}^T\|_F^2 - \mathbb{E}_{\mathbf{Y}'}(\|\mathbf{K} - \mathbf{Y}'\mathbf{Y}'^T\|_F^2),$$

where $\mathbf{Y}'$ denotes the one-hot encoded label matrix generated by the permutation of the true label vector $Y$ and $\mathbb{E}_{\mathbf{Y}'}$ is the expectation over the uniform distribution of $\mathbf{Y}'$.

According to the definition of PSED, $\mathbb{E}_{\mathbf{Y}'}(d_{SED}(\mathbf{K}, \mathbf{Y}'))$ requires the computation of all possible cases that follow the same distribution as the true label $Y$, involving a total of $\frac{n!}{m_1! m_2! \cdots m_k!}$ terms. As a result, its computational complexity is relatively high. To improve computational efficiency, an analytical solution for $\mathbb{E}_{\mathbf{Y}'}(d_{SED}(\mathbf{K}, \mathbf{Y}'))$ has been proposed:

**Theorem 3.2.** *Let $\mathbf{i}_r^n$ be the set of all $r$-tuples drawn without replacement from the set $\{1, \cdots, n\}$. The analytic solution of the expectation of $\mathbb{E}_{\mathbf{Y}'}(d_{SED}(\mathbf{K}, \mathbf{Y}'))$ is:*

$$\mathbb{E}_{\mathbf{Y}'}(d_{SED}(\mathbf{K}, \mathbf{Y}')) \qquad (6)$$
$$= \|\mathbf{K}\|_F^2 + \sum_{r=1}^{k} m_r^2 - 2\left( \sum_{i=1}^{n} \mathbf{K}_{ii} + \sum_{r=1}^{k} \frac{|\mathbf{i}_2^{m_r}|}{|\mathbf{i}_2^n|} \sum_{i,j;i\neq j}^{n} \mathbf{K}_{ij} \right),$$

*where $|\cdot|$ denotes the size of set.*

Based on Formula 29 , Formula 5 and Theorem 38, the analytic solution of PSED is:

$$d_{PSED}(\mathbf{K}) = \qquad (7)$$
$$2\left( \sum_{i=1}^{n} \mathbf{K}_{ii} + \sum_{r=1}^{k} \frac{|\mathbf{i}_2^{m_r}|}{|\mathbf{i}_2^n|} \sum_{i,j;i\neq j}^{n} \mathbf{K}_{ij} - \sum_{r=1}^{k} \mathbf{1}_{m_r} \mathbf{K}_{[r][r]} \mathbf{1}_{m_r} \right),$$

where $\mathbf{K}_{ij}$ is the value of the $i$-th row and $j$-th column of matrix $\mathbf{K}$. From the analytical expression, it is evident that the smaller the value of the expression, the closer the matrix $\mathbf{K}$ is to $\mathbf{K}_{[r][r]}$. This shows that in this case, the structure of $\mathbf{K}$ is closer to the class structure.

Next, we provide an analytical solution for PSED with a computational complexity of $\mathcal{O}(kn^2 + (1 - k)n + \sum_{i=1}^{k} m_i^2)$. Compared to the computational complexity of $\frac{n!}{m_1! m_2! \cdots m_k!} \times \mathcal{O}(2kn^2 + 2n^2)$ in Formula 5, the analytical solution significantly accelerates the computation speed of the expectation of PSED, ensuring that PSED can serve as an efficient computational objective function.

## 4. Properties Analysis

In this section, we mainly analyze the advantages of PSED compared to $d_{infor}$ and SED.

*Property* 1. (**Homogeneity of $d_{infor}$**) Suppose there are $n^2$ elements, let $\mathbf{K}$ and $\mathbf{K}'$ be two square matrices that are generated by arranging the $n^2$ elements in different ways. Then we have $d_{infor}(\mathbf{K}) = d_{infor}(\mathbf{K}')$.

*Property* 2. (**Heterogeneity of $d_{PSED}$**) Suppose there are $n^2 - n$ elements, let the diagonal positions of $\mathbf{K}$ and $\mathbf{K}'$ be 1 and their other positions are assigned by the $n^2 - n$ elements in different ways. Then if $d_{SED}(\mathbf{K}) \leq d_{SED}(\mathbf{K}')$, we have that $d_{PSED}(\mathbf{K}) < d_{PSED}(\mathbf{K}')$.

Properties 1 and 2 can be easily derived from the definition of $d_{infor}$ and Theorem 38, respectively. These properties

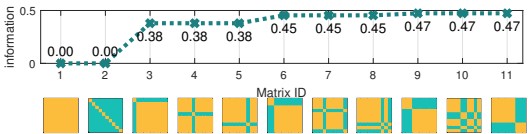

(a) The same $d_{infor}$ value for different adjacency matrices

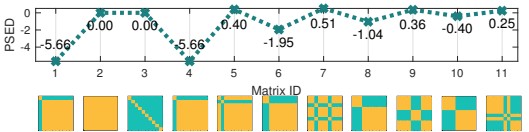

(b) The different PSED value for different adjacency matrices

*Figure 1.* Comparison with $d_{infor}$.

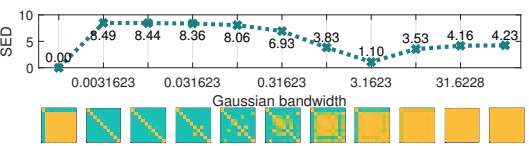

(a) The bias of SED in the imbalanced scene

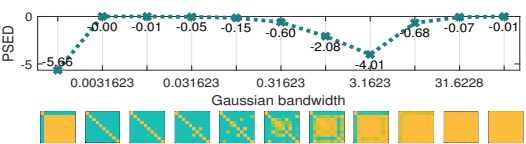

(b) The unbias of PSED in the imbalanced scene

*Figure 2.* Comparison with SED.

demonstrate that, compared to $d_{infor}$, PSED can effectively distinguish matrices with different internal structures.

We also verify the above properties in Figure 1 by providing three sets of adjacency matrices. Each set consists of matrices with identical proportions of zeros and ones but differing in their internal structural arrangements, as shown by ($\{3,4,5\},\{6,7,8\},\{9,10,11\}$). The identical $d_{infor}$ value for each set indicates that $d_{infor}$ cannot distinguish matrices with different internal structures. In contrast, from Figure 1(b), the $4,6,8,10$ matrices are closer to the first true adjacency matrix and exhibit lower PSED values. This phenomenon demonstrates the discrimination ability of PSED.

*Property* 3. (**Bias of** $d_{SED}$) For the matrices $I$ and $J$, when $\sum_{i=1}^{k} m_i^2 > \frac{n^2+n}{2}$, we have $d_{SED}(I) > d_{SED}(J)$; otherwise, $d_{SED}(I) < d_{SED}(J)$.

*Property* 4. (**Unbias of** $d_{PSED}$) For any matrix $A$ in $N_a = \{(1-a)I + aJ, 0 \leq a \leq 1\}$, where $I$ is the identity matrix and $J$ is the full one matrix. We have $d_{PSED}(A) = 0$.

From the above two properties, we conclude that SED is biased towards the non-informative matrices $I$ and $J$, whereas PSED assigns the same score to the non-informative simi-

larity matrices.

We also verify the above properties in Figure 2 with providing some Gaussian kernel matrices with different parameters. Figure 2 depicts SED and PSED values between the target matrix (the first one) and the kernel matrices. From Figure 2(a), we observe that SED is bias to the identity matrix. From Figure 2(b), both the second diagonal and the last full one matrix obtain the highest score. This signifies that PSED is unbias to any non informative matrix. The above advantages ensure that PSED is more appropriate to measure the quality of similarity matrix.

## 5. Learn ability of $d_{PSED}$ Loss

In machine learning, the underlying probability distribution of $\mathcal{X} \times \mathcal{Y}$ is usually unknown. Only a collection of empirical data $\mathcal{S}_n = \{(\boldsymbol{x}_1, y_1), ..., (\boldsymbol{x}_n, y_n)\}$ is available. Based on these empirical data, the $d_{PSED}$ is estimated by:

$$\hat{d}_{PSED}(\hat{\mathbf{K}}) = 2\left( \sum_{i=1}^{n} \hat{\mathbf{K}}_{ii} + \sum_{r=1}^{k} \frac{|\mathbf{i}_2^{m_r}|}{|\mathbf{i}_2^n|} \sum_{i,j;i\neq j}^{n} \hat{\mathbf{K}}_{ij} \quad (8) \\ - \sum_{r=1}^{k} \mathbf{1}_{m_r} \hat{\mathbf{K}}_{[r][r]} \mathbf{1}_{m_r} \right),$$

where $\hat{\mathbf{K}}_{ij} = \mathbf{Ker}(h(\boldsymbol{x}_i), h(\boldsymbol{x}_j))$ is a kernel function, $h \in \mathcal{H}$ is the hypothesis function that outputs the embedding representation vector, $\hat{\mathbf{K}}_{[r][r]}$ is the sub kernel matrix of class $r$ and $m_r$ is the number of objects of $r$ class.

By minimizing the empirical $\hat{d}_{PSED}$, a currently optimal classifier can be obtained. The generalization ability of this classifier can be characterized by the quality of the convergence of empirical loss to the true one. Due to the randomness of samples, the convergence is analyzed in terms of probability. Formally, let $\epsilon > 0$, the convergence analysis aims to find upper bound $\delta(\epsilon)$ on the probability of deviation inequalities:

$$\mathbb{P}(|d_{PSED}(\mathbf{K}) - \hat{d}_{PSED}(\hat{\mathbf{K}})| \geq \epsilon) \leq \delta(\epsilon), \quad (9)$$

where $\mathbf{K}_{ij} = \mathbb{E}_{\boldsymbol{X}_i, \boldsymbol{X}_j} \mathbf{Ker}(h(\boldsymbol{X}_i), h(\boldsymbol{X}_j))$ is the expectation of the kernel function.

The probability upper bound quantifies how quickly and accurately an empirical measure approaches the true measure as the sample size increases. Additionally, it provides insights into how the model structure and complexity affect the convergence performance. To establish the generalization ability bound, we employ the exponential Orlicz norm-based concentration inequality.

## 5.1. Exponential Orlicz Norm-based Concentration Inequality

Concentration inequalities (Boucheron et al., 2013) provide bounds on the probability that a random variable deviates from its mean or median. These inequalities are powerful for understanding the behavior of random processes, particularly in machine learning, where they help analyze a model's generalization ability and stability. Traditional inequalities, such as Markov's inequality, often yield loose bounds. By utilizing the bounded difference property, tighter inequalities, such as McDiarmid's or Hoeffding's inequalities, can be derived. Recently, an inequality based on the exponential Orlicz norm has been proposed (Escande, 2024).

### 5.1.1. EXPONENTIAL ORLICZ NORM

For $q \geq 1$, the $q$-exponential Orlicz norm of a random variable $X$ on the probability space $(\mathbb{X}, \mu)$ is defined as:

$$\|X\|_{\psi_q} = \inf_{c>0}\{\mathbb{E}\left[\exp\left(|X/c|^q\right)\right] \leq 2\}. \quad (10)$$

When $q = 1$ and $q = 2$, the norm are corresponds to sub-exponential and exponential Orlicz norms, respectively. When $\boldsymbol{X} \in \mathbb{R}^d$ is a random vector, its $\psi_q$ norm is defined by $\|\boldsymbol{X}\|_{\psi_q} = \sup_{v \in \mathbb{S}^{d-1}} \|\langle \boldsymbol{X}, v\rangle\|_{\psi_q}$, where $\mathbb{S}^{d-1}$ is the unit ball in $\mathbb{R}^d$ space. Next, we list three properties:

*Property* 5. Let $X$ and $Y$ be random variables, we have,

$$\|X + Y\|_{\psi_1} \leq 2(\|X\|_{\psi_1} + \|Y\|_{\psi_1}). \quad (11)$$

*Property* 6. Let $X_i, i = 1, .., L$ be random variables, we have,

$$\left\|\prod_{i=1}^{L} X_i\right\|_{\psi_1} \leq \prod_{i=1}^{L} \|X_i\|_{\psi_L}. \quad (12)$$

*Property* 7. Let $\boldsymbol{X} \in \mathcal{R}^d$ be random vector, we have,

$$\left\|\|\boldsymbol{X}\|_1\right\|_{\psi_q} \leq \sqrt{d}\|\boldsymbol{X}\|_{\psi_q}. \quad (13)$$

### 5.1.2. CONCENTRATION INEQUALITY

The inequality based on $\|X\|_{\psi_q}$ offers sharper bounds and provides a generalization performance bound of order $\mathcal{O}(1/n)$, where $n$ is the sample size.

**Theorem 5.1.** *(Escande, 2024) Let $f : \mathcal{X}^n \rightarrow \mathbb{R}$ and $\mathscr{B} \in \mathcal{X}^n$ such that $p = \mathbb{P}(X^n \notin \mathscr{B}) \leq 3/4$. For any two samples with only one different observation: $\mathcal{S}_n = \{\boldsymbol{x}_1, ..., \boldsymbol{x}_{k-1}, \boldsymbol{x}_k, ..., \boldsymbol{x}_n\}$ and $\mathcal{S}_{n,k} = \{\boldsymbol{x}_1, ..., \boldsymbol{x}_{k-1}, \boldsymbol{x}'_k, ..., \boldsymbol{x}_n\}$, assume there exist a pseudo metric $b : \mathcal{X} \times \mathcal{X} \rightarrow \mathbb{R}^+$ with $\|b\|_{\psi_1} < +\infty$ such that:*

$$|f(\mathcal{S}_n) - f(\mathcal{S}_{n,k})| \leq b(\boldsymbol{x}_k, \boldsymbol{x}'_k).$$

*Then with probability at least $1 - 2(\rho + \delta)$, where $\delta > 0$, we have,*

$$|f(\boldsymbol{X}_1, ..., \boldsymbol{X}_n) - \mathbb{E}\left[f \mid (\boldsymbol{X}_1, ..., \boldsymbol{X}_n) \in \mathscr{B}\right]|$$
$$\leq 4n\|b\|_{\psi_1}\sqrt{p} + e\|b\|_{\psi_1}\left(2\sqrt{n\log\left(\frac{1}{\delta}\right)} + \log\left(\frac{1}{\delta}\right)\right).$$

Theorem 5.1 states that if one sample among the $n$ objects is modified, the change in the function value over all $n$ objects is bounded by the magnitude of the change in the individual sample. Consequently, the deviation between the function value and its expectation can be characterized by the exponential Orlicz norm of the change magnitude.

## 5.2. Network Structure

This paper considers fully connected layer networks to obtain representation vectors. Given $L$ weight matrices $\boldsymbol{W} = (\boldsymbol{W}_1, ..., \boldsymbol{W}_L)$ and $L$ activation functions $(\boldsymbol{\sigma}_1, ..., \boldsymbol{\sigma}_L)$, where $\boldsymbol{\sigma}_i : \mathbb{R}^{d_{i-1}} \rightarrow \mathbb{R}^{d_i}$ and $d_i$ is the output dimension of $i$-th layer. The fully connected network $h_{\mathcal{W},\mathcal{L}}$ is:

$$h_{\mathcal{W},L}(\boldsymbol{x}) := \boldsymbol{\sigma}_L(\boldsymbol{W}_L\boldsymbol{\sigma}_{L-1}(\boldsymbol{W}_{L-1}\cdots\boldsymbol{\sigma}_1(\boldsymbol{W}_1\boldsymbol{x}))), \quad (14)$$

where $\boldsymbol{\sigma}$ is the nonlinear activation function. The common used activation functions, coordinate-wise ReLU and sigmoid function, are $\rho_i$-Lipschitz continuous (Bartlett et al., 2017). The $\rho_i$-Lipschitz continuous requires that for all $\boldsymbol{z}$, $\boldsymbol{z}'$ in its domain, the following inequality holds:

$$\|\boldsymbol{\sigma}_i(\boldsymbol{z}) - \boldsymbol{\sigma}_i(\boldsymbol{z}')\|_p \leq \rho_i\|\boldsymbol{z} - \boldsymbol{z}'\|_p, \quad (15)$$

where $\|\cdot\|_p$ is the $p$-norm. This ensures that the function does not change too rapidly, with $\rho_i$ serving as the Lipschitz constant that bounds the growth.

With the Lipschitz continuity, the output variation of a fully connected network can be bounded by the sample perturbation. For two observations $\boldsymbol{x}_k$ and $\boldsymbol{x}'_k$, their output satisfies:

$$\|h_{\mathcal{W},i}(\boldsymbol{x}_k) - h_{\mathcal{W},i}(\boldsymbol{x}'_k)\|_p \leq \|\boldsymbol{x}_k - \boldsymbol{x}'_k\|_p \prod_{j=1}^{i-1} \rho_j\|W_j\|_p. \quad (16)$$

The reason is that based on the Lipschitz continuity, for two observations $\boldsymbol{x}_k$ and $\boldsymbol{x}'_k$, we have:

$$\|h_{\mathcal{W},i}(\boldsymbol{x}_k) - h_{\mathcal{W},i}(\boldsymbol{x}'_k)\|_p \quad (17)$$
$$= \|\sigma_i(W_ih_{\mathcal{W},i-1}(\boldsymbol{x}_k)) - \sigma_i(W_ih_{\mathcal{W},i-1}(\boldsymbol{x}'_k))\|_p \quad (18)$$
$$\leq \rho_i\|W_ih_{\mathcal{W},i-1}(\boldsymbol{x}_k) - W_ih_{\mathcal{W},i-1}(\boldsymbol{x}'_k)\|_p \quad (19)$$
$$\leq \rho_i\|W_i\|_p\|h_{\mathcal{W},i-1}(\boldsymbol{x}_k) - h_{\mathcal{W},i-1}(\boldsymbol{x}'_k)\|_p \quad (20)$$

where the last inequality is based on the Cauchy Schwartz inequality. Further, by successive application of this property, we can obtain Eq. (16).

## 5.3. Generalization Bound for $d_{PSED}$ Loss

Based Theorem 5.1, our bound is:

**Theorem 5.2.** *When the hypothesis function is the fully connected layer networks and* $\mathbf{K}$ *is the RBF kernel* $K(x_i, x_j) = \exp\left(-\gamma(x_i - x_j)^2\right)$. *Let* $\mathcal{B} \in \mathcal{X}^n$ *such that* $p = \mathbb{P}(\boldsymbol{X}^n \notin \mathcal{B}) \leq 3/4$, *for* $\delta > 0$, *Then with probability at least* $1 - 2(\rho + \delta)$, *we have,*

$$\left| d_{PSED}(\mathbf{K}) - \hat{d}_{PSED}(\hat{\mathbf{K}}) | \boldsymbol{X} \in \mathcal{B} \right|$$

$$\leq \|b\|_{\psi_1} \left( 4\sqrt{p} + e\left( 2\sqrt{\frac{1}{n}\log\left(\frac{1}{\delta}\right)} + \frac{1}{n}\log\left(\frac{1}{\delta}\right) \right) \right),$$

*where,*

$$\|b\|_{\psi_1} = \left(2 + 4C(n-1) + 4\max\{m_r\}_{r=1}^k\right) \quad (21)$$

$$2\gamma M\sqrt{d}\|diam(\boldsymbol{x})\|_{\psi_L} \prod_{l=1}^{L-1} \rho_l \sqrt{d_l}\|\boldsymbol{W}_l\|_{\psi_L},$$

$C = \sum_{r=1}^k \frac{|\mathbf{i}_2^{m_r}|}{|\mathbf{i}_2^n|}$ *and* $M = 2\max_{\boldsymbol{x} \in \mathcal{X}} \|\boldsymbol{x}\|$.

From Theorem 5.2, we can conclude that, for the fully connected network, the generalization bound is related to the following terms: $\|diam(\boldsymbol{x})\|_{\psi_L}$ is the exponential Orlicz norms of the input domain diameter; $d_j$ is the number of nodes in each layer of the network, and $d$ is the input dimensional; $\rho_l$ is the Lipschitz constant of $l$-th activation function; $\|\boldsymbol{W}_l\|_{\psi_L}$ is the exponential Orlicz norms of $l$-th weight vector; $n$ is the number of training instances. From Theorem 5.2, the smaller the exponential Orlicz norms of the input domain diameter and the network parameter vector, the fewer the network nodes and the more the samples, the smaller the model's generalization error.

# 6. Methodology

This paper presents a deep learning framework designed to learn discriminative feature representations through a novel loss function that serves as a universal similarity matrix quality measure, applicable across diverse learning paradigms including deep network training, metric learning, and kernel methods. To demonstrate its versatility, we implement the approach on three fundamental architectures: fully connected networks (whose schematic diagram is shown in Figure 3), Vision Transformers (ViT), and Convolutional Neural Networks (CNN), with experimental details for ViT and CNN provided in the Appendix.

The framework operates by first transforming input data into latent embeddings, computing pairwise similarity matrices in the feature space, then optimizing network parameters through backpropagation using our proposed debiased distance metric that effectively evaluates representation quality while overcoming limitations of conventional similarity

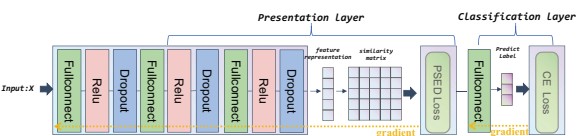

*Figure 3.* The framework of the proposed method on fully connected networks.

measures. This generalized formulation maintains theoretical rigor while enabling practical applications across multiple deep learning architectures.

**Presentation layer (PL)**

The model consists of three layers, each containing a fully connected layer, a ReLU activation layer, and a Dropout regularization layer. The fully connected layer performs a linear transformation of the previous layer's activations through the weight matrix $\mathbf{W}$, producing new feature representations. The ReLU activation introduces nonlinearity, enhancing the model's ability to capture complex patterns. The Dropout layer randomly drops neurons to prevent overfitting. Subsequently, the similarity matrix is computed by taking the inner product of the hidden layer activations. Finally, the processed features are optimized using the PSED loss function, with backpropagation applied to minimize the prediction error. The relevant formula is as follows:

$$\boldsymbol{Z}^l = \boldsymbol{W}_{l-1}\boldsymbol{Z}^{l-1}, \quad (22)$$

$$\boldsymbol{Z}^l = ReLU(\boldsymbol{Z}^l) = \max(0, \boldsymbol{Z}^l), \quad (23)$$

$$\boldsymbol{Z}^l = Dropout(\boldsymbol{Z}^l), \quad (24)$$

$$\mathbf{K} = \boldsymbol{Z}^l(\boldsymbol{Z}^l)^T, \quad (25)$$

where $\boldsymbol{W}_{l-1}$ represent the parameters of the $(l-1)$th fully connected layer.

**Classification layer (CL)**

The prediction labels $\hat{Y}$ are generated through additional fully connected layers:

$$\tilde{Y} = \boldsymbol{W}_{l+1}\boldsymbol{Z}^l, \quad \hat{Y} = softmax(\tilde{Y}). \quad (26)$$

To quantify the discrepancy between the predicted probability distribution and the true distribution, the cross-entropy (CE) loss function is used. The CE is defined as:

$$\text{CE} = -\sum_{i=1}^n \sum_{j=1}^k Y_{ij} \log \tilde{Y}_{ij}, \quad (27)$$

where $n$ is the number of samples, $k$ is the number of classes, $\tilde{Y}_{ij}$ is the predicted probability that sample $i$ belongs to class $j$ and $Y_{ij}$ is the true label. The loss value is minimized using the back propagation algorithm, which optimizes the parameters of the final fully connected layer.

# 7. Experiment

In this section, we compare our proposed method with three common loss functions on 20 benchmark datasets and 5 image datasets. And we compare the methods based on CE, SED, $1\text{-}d_{infor}(\mathbf{K})$, and PSED loss functions at the presentation layer. Additionally, we conduct analysis experiments to further demonstrate the advantages of our method.

## 7.1. Experimental on Benchmark Dataset

### 7.1.1. PERFORMANCE ANALYSIS

In this section, we report the average accuracy and F-measure of four methods on benchmark datasets with 10 partitions and 3 model layers. Table 1 shows the results, with each row representing a dataset and columns divided by evaluation metric. The highest value in each section is bolded. If our method significantly outperforms the others, a black dot will be placed next to the method (for significance testing, please refer to the Appendix). As indicated by the table, the PSED-based loss function demonstrates superior performance in terms of average convergence accuracy and F-measure, with values surpassing those of other methods on most datasets.

### 7.1.2. SIGNIFICANCE TEST

To demonstrate the superiority of CE-PSED, we first conduct a Friedman test to confirm significant differences among the methods, followed by a Nemenyi post-hoc test to identify specific pairs with differences (details for layers 5 and 8 are in the Appendix). Figure 4 shows the CD diagram for 3-layer models, where the x-axis represents the average rank and the CD line indicates the critical difference from the Nemenyi test. Methods marked with a red star indicate the best performance, while those not connected by a red line show significant performance differences. As shown in Figure 4, the CE-PSED-based method has a significantly lower average rank. Not only does CE-PSED achieve the best performance, indicated by the red star, but it is also not connected by a red line to any other method. This indicates that its accuracy and F-measures are superior to those of other algorithms across multiple datasets.

Additionally, a further significance test was conducted to validate the enhanced performance of CE-PSED, with methodological details available in (Wang et al., 2023; Li et al., 2019). Figure 5 presents the significance test results for all datasets, where each bar chart illustrates the difference between the number of times the algorithm's significance wins and losses. As shown in Figure 5, the PSED-based method significantly outperforms the other methods.

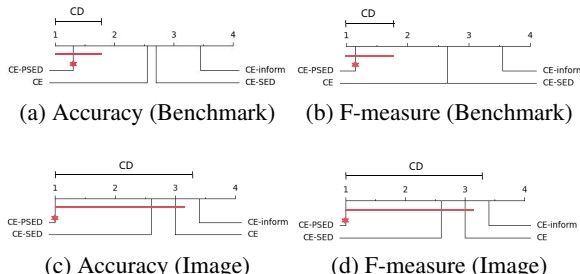

(a) Accuracy (Benchmark)   (b) F-measure (Benchmark)

(c) Accuracy (Image)   (d) F-measure (Image)

*Figure 4.* CD diagrams w.r.t. Accuracy and F-measure.

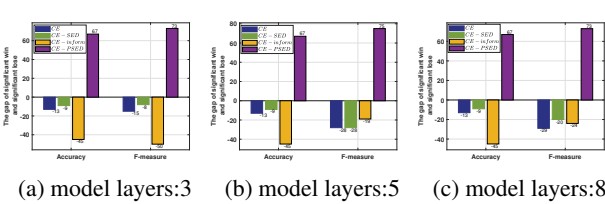

(a) model layers:3   (b) model layers:5   (c) model layers:8

*Figure 5.* Significance test w.r.t. Accuracy and F-measure.

### 7.1.3. CONVERGENCE ANALYSIS

The Figure 6 (a) and (b) shows the performance of the four methods over the training epoch in the benchmark datasets with the highest and lowest degree of imbalance (see appendix for other dataset results), where the points on each line represent the average accuracy of the corresponding period. The results show that the CE-PSED method exhibits significant performance advantages in most datasets and training epochs, and can quickly converge, which fully demonstrates its robustness and effectiveness on datasets.

### 7.1.4. NETWORK LAYER ANALYSIS

To verify the effectiveness of the CE-PSED method at different network layers, this study set the model layers to 5 and 8, respectively, and selected one dataset for presentation (see Appendix for other datasets). Tables 3 presents the F-measure values of four methods at different levels. The results show that as the number of network layers increased, the F-measure values of other methods significantly decreased, while the F-measure value of the CE-PSED method decrease less and remain the highest. This indicates that other models experience feature representation collapse as the number of layers increases, that is, the model tends to classify samples into the same category, while the CE-PSED method performs well at different layers, effectively avoiding feature collapse and ensuring that the model maintains good feature learning and classification capabilities.

*Table 1.* Accuracy and F-measure based on different loss functions on the benchmark datasets when model layers is 3.

| Data | Accuracy | | | | F-measure | | | |
|---|---|---|---|---|---|---|---|---|
| | CE | CE-SED | CE-inform | CE-PSED | CE | CE-SED | CE-inform | CE-PSED |
| 1 | 0.6796±0.1049 | 0.8900±0.0104 | 0.7869±0.0878 | **0.9118±0.0001** | 0.6840±0.0995● | 0.8727±0.0034 | 0.7703±0.0716 | **0.8965±0.0001** |
| 2 | 0.7662±0.0000● | 0.7702±0.0001● | 0.7920±0.0001 | **0.8049±0.0002** | 0.6758±0.0001 ● | 0.6865±0.0007● | 0.7490±0.0006 ● | **0.7806±0.0004** |
| 3 | 0.7558±0.0002 | 0.7667±0.0005 | 0.7662±0.0006 | **0.7723±0.0007** | 0.7486±0.0002 | 0.7602±0.0005 | 0.7623±0.0006 | **0.7686±0.0007** |
| 4 | 0.9219±0.0006 | 0.9219±0.0003 | 0.6535±0.0000 ● | **0.9278±0.0002** | 0.9200±0.0007 | 0.9204±0.0004 | 0.5204±0.0004 ● | **0.9264±0.0002** |
| 5 | 0.8313±0.0003 | 0.8362±0.0003 | 0.6707±0.0000 ● | **0.8423±0.0005** | 0.8306±0.0003 | 0.8354±0.0003 | 0.5393±0.0000● | **0.8424±0.0005** |
| 6 | 0.8662±0.0003 | 0.8700±0.0002 | 0.6767±0.0015 ● | **0.8707±0.0003** | 0.8661±0.0003 | 0.8698±0.0002 | 0.5760±0.0062 ● | **0.8706±0.0003** |
| 7 | 0.5552±0.0004 ● | 0.5646±0.0008● | 0.5686±0.0004● | **0.5993±0.0004** | 0.5475±0.0005● | 0.5556±0.0008● | 0.5470±0.0006● | **0.5903±0.0004** |
| 8 | 0.9157±0.0002 | 0.8841±0.0002 ● | 0.4441±0.0017● | **0.9165±0.0001** | 0.9157±0.0002● | 0.8840±0.0002● | 0.3738±0.0027 | **0.9166±0.0001** |
| 9 | 0.6864±0.0002● | 0.6175±0.0011 ● | 0.4775±0.0022● | **0.7389±0.0003** | 0.6759±0.0003● | 0.5811±0.0015● | 0.3598±0.0034 ● | **0.7382±0.0002** |
| 10 | 0.9212±0.0001 | 0.9191±0.0001 ● | 0.8489±0.0000 ● | **0.9347±0.0001** | 0.9200±0.0001 | 0.9184±0.0001 ● | 0.7795±0.0000 ● | **0.9334±0.0001** |
| 11 | **0.9711±0.0000** | **0.9711±0.0000** | **0.9711±0.0000** | 0.9607±0.0000 | 0.9568±0.0000 | 0.9568±0.0000 | 0.9568±0.0000 | 0.9603±0.0000 |
| 12 | 0.7783±0.0229 | 0.7986±0.0132 | 0.6535±0.0207 ● | **0.8469±0.0015** | 0.7420±0.0482 | 0.7762±0.0269 | 0.5789±0.0449● | **0.8430±0.0017** |
| 13 | 0.9763±0.0000 ● | 0.9760±0.0000 ● | 0.9474±0.0000 ● | **0.9838±0.0000** | 0.9753±0.0000 ● | 0.9754±0.0000 ● | 0.9218±0.0000● | **0.9835±0.0000** |
| 14 | 0.9831±0.0000● | 0.9831±0.0000● | 0.9484±0.0000 ● | **0.9899±0.0000** | 0.9825±0.0000 ● | 0.9825±0.0000 ● | 0.9232±0.0000● | **0.9898±0.0000** |
| 15 | 0.6711±0.0002● | 0.6830±0.0003 | 0.6113±0.0003 ● | **0.6930±0.0001** | 0.6710±0.0002● | 0.6831±0.0003 | 0.6089±0.0004 ● | **0.6927±0.0001** |
| 16 | 0.9403±0.0001 | 0.9397±0.0001 | **0.9628±0.0001** | 0.9495±0.0003 | 0.9130±0.0004 | 0.9118±0.0003 | **0.9552±0.0003** | 0.9299±0.0009 |
| 17 | 0.9266±0.0000● | 0.9185±0.0001● | 0.9217±0.0000 ● | **0.9732±0.0000** | 0.9266±0.0000● | 0.9185±0.0001● | 0.9217±0.0000● | **0.9732±0.0000** |
| 18 | 1.0000±0.0000 | 1.0000±0.0000 | 0.9832±0.0000 ● | **1.0000±0.0000** | 1.0000±0.0000 | 1.0000±0.0000 | 0.9832±0.0000● | **1.0000±0.0000** |
| 19 | 0.9532±0.0000● | 0.9510±0.0000● | 0.9033±0.0000● | **0.9956±0.0000** | 0.9526±0.0000● | 0.9505±0.0001● | 0.8924±0.0000 ● | **0.9955±0.0000** |
| 20 | 0.8453±0.0000 | 0.8445±0.0000 | 0.8436±0.0000● | **0.8454±0.0000** | 0.8399±0.0000 ● | 0.8390±0.0000 | 0.8381±0.0000 | **0.8625±0.0000** |

*Table 2.* Accuracy and F-measure based on different loss functions on the image datasets when model layers is 3.

| Data | Accuracy | | | | F-measure | | | |
|---|---|---|---|---|---|---|---|---|
| | CE | CE-SED | CE-inform | CE-PSED | CE | CE-SED | CE-inform | CE-PSED |
| Mpeg | 0.6438±0.0001● | 0.6538±0.0004 ● | 0.2098±0.0006 ● | **0.7338±0.0002** | 0.6342±0.0001● | 0.6426±0.0004● | 0.1549±0.0004 ● | **0.7244±0.0003** |
| Mnist | 0.9039±0.0000 | 0.9046±0.0000 | 0.8423±0.0001 ● | **0.9062±0.0000** | 0.9037±0.0000 | 0.9044±0.0000 | 0.8409±0.0001● | **0.9061±0.0000** |
| Pendigits | 0.7761±0.0009● | 0.9611±0.0000● | 0.7889±0.0010● | **0.9908±0.0000** | 0.7684±0.0011● | 0.9610±0.0000● | 0.7808±0.0013● | **0.9908±0.0000** |
| Caltech-101 | 0.3561±0.0001● | 0.2778±0.0002● | 0.2758±0.0001 ● | **0.5005±0.0001** | 0.2814±0.0001● | 0.1786±0.0003 ● | 0.1754±0.0002 ● | **0.4708±0.0001** |
| ImageNet | 0.9701±0.0000 ● | 0.9699±0.0000● | 0.9753±0.0000 | **0.9762±0.0000** | 0.9703±0.0000● | 0.9701±0.0000● | 0.9754±0.0000 | **0.9762±0.0000** |

*Table 3.* Comparison of F-measure at different depths on Yeast

| Method | Layer 3 | Layer 5 | Layer 8 |
|---|---|---|---|
| CE | 0.5475 | 0.1539 | 0.1662 |
| CE-SED | 0.5556 | 0.1734 | 0.1740 |
| CE-inform | 0.5470 | 0.1620 | 0.1646 |
| CE-PSED | **0.5903** | **0.4951** | **0.4116** |

### 7.1.5. ANALYSIS OF DISCERNMENT ABILITY

To evaluate the discriminative ability of the four methods, we analyze the feature representations of the last hidden layer of the model. Figure 16 illustrates the t-SNE visualization of these feature representations for each method on the Pendigits dataset (see Appendix for results on other datasets). The figure demonstrates that the PSED-based method effectively separates classes. Additionally, we compute the Euclidean distances and information entropies between the similarity matrices and $YY^T$ across all benchmark and image datasets. For the Pendigits dataset, the Euclidean distances for CE, CE-SED, CE-inform, and CE-PSED are 1055.0093, 1055.0093, 1042.0280, and 919.8159, respectively, while the corresponding information entropies are

837.9047, 837.9047, 837.8022, and 837.8456 (see Appendix for additional results). These findings further validate the superior performance of CE-PSED in feature representation and class discrimination.

### 7.2. Experimental on the Image Dataset

We also evaluate the proposed method on five additional image datasets to further validate its effectiveness. These experiments adhere to the same settings as the baseline dataset, differing only in the feature extraction methods (see Appendix for details), ensuring consistency and comparability of the results. As shown in Table 2 and Figures 6(c) and (d), the results clearly demonstrate that the CE-PSED-based method outperforms other methods in both performance and efficiency in recognizing category structures. Specifically, the CE-PSED method excels in multiple key performance metrics, including classification accuracy and F-measure. Additionally, it exhibits a particularly strong capability in revealing structural differences between categories, a critical aspect of image recognition and classification tasks.

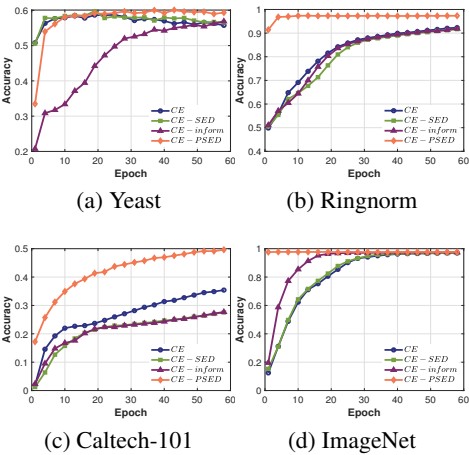

*Figure 6.* Accuracy curves when model layers is 3.

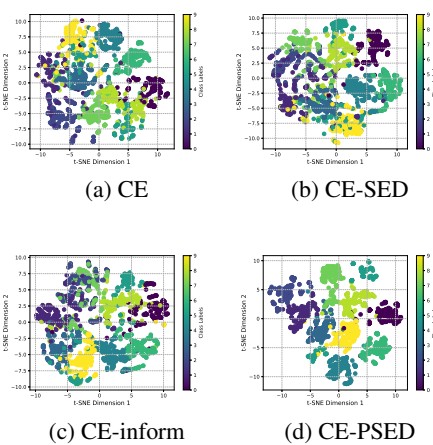

*Figure 7.* The t-SNE of Pendigits.

## 8. Conclusion

This paper introduces a novel Pure Square Euclidean Distance (PSED) metric within the framework of pure random consistency and provides a corresponding analytical solution. The unbiasedness and heterogeneity of PSED are rigorously validated through both theoretical analysis and simulation experiments. Additionally, the study investigates the learnability of PSED in fully connected neural network structures and establishes its performance. Furthermore, we propose a deep network model that utilizes PSED as the loss function, demonstrating superior performance and effectively mitigating collapse. In the future, we plan to analyze the optimization convergence properties of PSED and develop further learning models that optimize PSED.

## Acknowledgements

This work was supported by the National Natural Science Foundation of China (Nos. 62306170, 62136005, U24A20253, 62476160, 62441239), the Major Project of National Natural Science Foundation of China (No. T2495251), the Science and Technology Major Project of Shanxi (No. 202201020101006), the Special Fund for Science and Technology Innovation Teams of Shanxi Province (No. 202304051001001).

## Impact Statement

This paper presents work whose goal is to advance the field of Machine Learning. There are many potential societal consequences of our work, none which we feel must be specifically highlighted here.

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

# Appendix

# 9. Proof

### 9.1. Proof of the analytical solution of SED

The Square Euclidean Distance (SED) is defined as:

$$d_{SED}(\mathbf{K}) = \|\mathbf{K} - YY^T\|_F^2, \tag{28}$$

where $Y$ is the real label vector, $Y \in \{0,1\}^{n \times k}$ is the one-hot encoding of the true label vector, $n$ is the number of instances, $k$ is the number of classes and $\|\cdot\|_F^2$ is the Frobenius norm, which represents the square of the sum of squared elements of the matrix.

And we provide an analytical solution for SED:

$$\begin{aligned} d_{SED}(\mathbf{K}, Y) &= \|\mathbf{K} - YY^T\|_F^2 \\ &= \|\mathbf{K}\|_F^2 + \sum_{i=1}^{k} m_i^2 - 2\sum_{i=1}^{k} \mathbf{1}_{m_i}^T \mathbf{K}_{[i][i]} \mathbf{1}_{m_i} \end{aligned} \tag{29}$$

where $\mathbf{1}_{m_i}$ is single column all 1 vectors of length $m_i$, $\mathbf{K}_{[i][i]}$ is the sub kernel matrix of class $i$ and $m_i$ is the number of objects of $i$ class.

**Proof** For multi-class classification tasks, let $n$ be the total number of samples, $k$ be the number of categories, $m_1, m_2, \cdots, m_k$ be the number of samples in each category, such that $m_1 + m_2 + \cdots + m_k = n$. The true label matrix $Y$ can be represented as:

$$Y = \begin{bmatrix} \mathbf{1}_{m_1 \times 1}, & \mathbf{0}_{m_1 \times 1}, & \cdots & \mathbf{0}_{m_1 \times 1} \\ \mathbf{0}_{m_2 \times 1}, & \mathbf{1}_{m_2 \times 1}, & \cdots & \mathbf{0}_{m_2 \times 1} \\ \vdots & \vdots & \vdots & \vdots \\ \mathbf{0}_{m_k \times 1}, & \mathbf{0}_{m_k \times 1}, & \cdots & \mathbf{1}_{m_k \times 1} \end{bmatrix}_{n \times k}^T, \tag{30}$$

where $\mathbf{1}_{m_1 \times 1}$ and $\mathbf{0}_{m_1 \times 1}$ are single column all 1 vectors and all 0 vectors of length $m_1$, respectively.

The adjacency matrix generated by $Y$ is:

$$YY^T = \begin{bmatrix} \mathbf{J}_{m_1 \times m_1} & \mathbf{0}_{m_1 \times m_2} & \cdots & \mathbf{0}_{m_1 \times m_k} \\ \mathbf{0}_{m_2 \times m_1} & \mathbf{J}_{m_2 \times m_2} & \cdots & \mathbf{0}_{m_2 \times m_k} \\ \vdots & \vdots & \vdots & \vdots \\ \mathbf{0}_{m_k \times m_1} & \mathbf{0}_{m_k \times m_2} & \cdots & \mathbf{J}_{m_k \times m_k} \end{bmatrix}_{n \times n}, \tag{31}$$

where $\mathbf{J}$ and $\mathbf{0}$ are the full one matrix and the full zero matrix, respectively. According to the category of samples, we block the similarity matrix as follows:

$$\mathbf{K} = \begin{bmatrix} \mathbf{K}_{[1][1]} & \mathbf{K}_{[1][2]} & \cdots & \mathbf{K}_{[1][k]} \\ \mathbf{K}_{[2][1]} & \mathbf{K}_{[2][2]} & \cdots & \mathbf{K}_{[2][k]} \\ \vdots & \vdots & \vdots & \vdots \\ \mathbf{K}_{[k][1]} & \mathbf{K}_{[k][2]} & \cdots & \mathbf{K}_{[k][k]} \end{bmatrix}_{n \times n}. \tag{32}$$

where $\mathbf{K}_{[i][j]}$ represents the similarity matrix between class $m_i$ and class $m_j$.

For $1 \le i = j \le k$,

$$\begin{aligned} &\|\mathbf{K}_{[i][i]} - YY_{m_i \times m_i}^T\|_F^2 \\ =&\|\mathbf{K}_{[i][i]}\|_F^2 - 2\langle \mathbf{K}_{[i][i]}, \mathbf{J}_{m_i \times m_i}\rangle + \|\mathbf{J}_{m_i \times m_i}\|_F^2 \\ =&\|\mathbf{K}_{[i][i]}\|_F^2 - 2\mathbf{1}_{m_i}^T \mathbf{K}_{[i][i]} \mathbf{1}_{m_i} + m_i^2. \end{aligned} \tag{33}$$

Similarly, for $1 \leq i \neq j \leq k$,

$$\|\mathbf{K}_{[i][j]} - \boldsymbol{Y}\boldsymbol{Y}_{m_i \times m_j}^T\|_F^2 \tag{34}$$

$$=\|\mathbf{K}_{[i][j]} - \boldsymbol{0}_{m_i \times m_j}\|_F^2$$

$$=\|\mathbf{K}_{[i][j]}\|_F^2. \tag{35}$$

Then by performing some simple elementary operations, we obtain:

$$d_{SED}(\mathbf{K}, \boldsymbol{Y}) = \|\mathbf{K} - \boldsymbol{Y}\boldsymbol{Y}^T\|_F^2 \tag{36}$$

$$= \|\mathbf{K}\|_F^2 + \sum_{i=1}^{k} m_i^2 - 2\sum_{i=1}^{k} \mathbf{1}_{m_i}^T \mathbf{K}_{[i][i]} \mathbf{1}_{m_i} \square$$

## 9.2. Proof of Theorem 3.2

To provide a proof for Theorem 3.2, we first give a lemma.

**Lemma 9.1.** *Let $A = \{a_1, a_2 \cdots a_n\}$ be a set of $n$ elements. Select $m$ elements from $A$, calculate the sum of the products of any $r$ elements selected from these $m$ elements as $S_1$ and compare it with the sum of the products of any $r$ elements selected from $A$ as $S_2$. Since they are only related to the number of items, the relationship between $\frac{S_1}{S_2}$ is:*

$$\frac{S_1}{S_2} = \frac{\sum_{l=1}^{|\mathbf{i}_m^n|} \sum_{v=1}^{|\mathbf{i}_r^m|} \prod_{i \in (\mathbf{i}_r^m)_v} a_i}{\sum_{l=1}^{|\mathbf{i}_r^n|} \prod_{i \in (\mathbf{i}_r^n)_l} a_i} = \frac{|\mathbf{i}_m^n| \times |\mathbf{i}_r^m|}{|\mathbf{i}_r^n|} \tag{37}$$

*where $(\mathbf{i}_m^n)_l$ represent the l-th set of $m$ elements taken from $n$ and $|\cdot|$ denotes the size of set.*

**Proof**

**Theorem 3.2** Let $\mathbf{i}_r^n$ be the set of all $r$-tuples drawn without replacement from the set $\{1, \cdots, n\}$. The analytic solution of the expectation of $\mathbb{E}_{\boldsymbol{Y}'}(d_{SED}(\mathbf{K}, \boldsymbol{Y}'))$ is:

$$\mathbb{E}_{\boldsymbol{Y}'}(d_{SED}(\mathbf{K}, \boldsymbol{Y}')) \tag{38}$$

$$= \|\mathbf{K}\|_F^2 + \sum_{i=1}^{k} m_i^2 - 2\left( \sum_{i=1}^{n} \mathbf{K}_{ii} + \sum_{r=1}^{k} \frac{|\mathbf{i}_2^{m_r}|}{|\mathbf{i}_2^n|} \sum_{i,j;i \neq j}^{n} \mathbf{K}_{ij} \right).$$

**Step 1: Convert the expectation about all permutation $Y'$ into the mean about the $m_1$-tuples, $m_2$-tuples,$\cdots$, $m_k$-tuples.**

According to the definition of PSED:

$$d_{PSED}(\mathbf{K}) = d_{SED}(\mathbf{K}, \boldsymbol{Y}) - \mathbb{E}_{\boldsymbol{Y}'}(d_{SED}(\mathbf{K}, \boldsymbol{Y}')) \tag{39}$$

$$= \|\mathbf{K} - \boldsymbol{Y}\boldsymbol{Y}^T\|_F^2 - \mathbb{E}_{\boldsymbol{Y}'}(\|\mathbf{K} - \boldsymbol{Y}'\boldsymbol{Y}'^T\|_F^2),$$

and the analytic solution of $d_{SED}$:

$$d_{SED}(\mathbf{K}, \boldsymbol{Y}) = \|\mathbf{K} - \boldsymbol{Y}\boldsymbol{Y}^T\|_F^2 \tag{40}$$

$$= \|\mathbf{K}\|_F^2 + \sum_{i=1}^{k} m_i^2 - 2\sum_{i=1}^{k} \mathbf{1}_{m_i}^T \mathbf{K}_{[i][i]} \mathbf{1}_{m_i}$$

we have,

$$\mathbb{E}_{\boldsymbol{Y}'}(d_{SED}(\mathbf{K}, \boldsymbol{Y}')) \tag{41}$$

$$= \|\mathbf{K}\|_F^2 + \sum_{i=1}^{k} m_i^2 - 2\sum_{i=1}^{k} \mathbb{E}_{\boldsymbol{Y}'}(\mathbf{1}_{m_i}^T \mathbf{K}'_{[i][i]} \mathbf{1}_{m_i})$$

where $\mathbf{K}'_{[i][i]}$ is the sub-block matrices of $\mathbf{K}'$.

The matrix $\mathbf{K}'$ is the permutation similarity matrix after switching the positions of the samples according to $\mathbf{Y}'$. Since $\mathbf{Y}'$ consists of all possible labels that maintain the distribution ratio $m_1 : m_2 : \cdots : m_k$, with the labels being uniformly distributed, the expectation can be expressed as:

$$
\sum_{r=1}^{k} \mathbb{E}_{\mathbf{Y}'}(\mathbf{1}_{m_i}^T \mathbf{K}'_{[r][r]} \mathbf{1}_{m_i}) \tag{42}
$$

$$
= \frac{\sum_{l=1}^{|\mathbf{i}_{m_1}^n|} \sum_{i,j \in (\mathbf{i}_{m_1}^n)_l} \mathbf{K}_{ij}}{|\mathbf{i}_{m_1}^n|} + \frac{\sum_{l=1}^{|\mathbf{i}_{m_2}^{n-m_1}|} \sum_{i,j \in (\mathbf{i}_{m_2}^{n-m_1})_l} \mathbf{K}_{ij}}{|\mathbf{i}_{m_1}^n| \times |\mathbf{i}_{m_2}^{n-m_1}|} + \cdots + \frac{\sum_{l=1}^{|\mathbf{i}_{m_k}^{n-m_1-\cdots-m_{k-1}}|} \sum_{i,j \in (\mathbf{i}_{m_k}^{n-m_1-\cdots-m_{k-1}})_l} \mathbf{K}_{ij}}{|\mathbf{i}_{m_1}^n| \times |\mathbf{i}_{m_2}^{n-m_1}| \times \cdots \times |\mathbf{i}_{m_k}^{n-m_1-\cdots-m_{k-1}}|}
$$

where $l \in \{1, 2, .., |\mathbf{i}_{m.}^n|\}$, $(\mathbf{i}_{m_1}^n)_l \cup (\mathbf{i}_{m_2}^{n-m_1})_l \cup \cdots \cup (\mathbf{i}_{m_k}^{n-m_1-\cdots-m_{k-1}})_l = \{1, \cdots, n\}$, $(\mathbf{i}_{m_1}^n)_l \cap (\mathbf{i}_{m_2}^{n-m_1})_l \cap \cdots \cap (\mathbf{i}_{m_k}^{n-m_1-\cdots-m_{k-1}})_l = \emptyset$, $n - m_1 - \cdots - m_{j-1}$ in $(\mathbf{i}_{m_j}^{n-m_1-\cdots-m_{j-1}})_l$ represents the set $\{1, 2, \cdots, n\} \backslash \{(\mathbf{i}_{m_1}^n)_l \cup (\mathbf{i}_{m_2}^{n-m_1})_l \cup \cdots \cup (\mathbf{i}_{m_{j-1}}^{n-m_1-\cdots-m_{j-2}})_l\}$, and $\mathbf{K}_{ij}$ is the value of the i-th row and j-th column of matrix $\mathbf{K}$.

**Step 2: Convert the mean about tuples into the mean of elements in K.**

Based on the observation, Formula 42 can be further computed using Lemma 9.1. In the lemma, when $r = 1$:

$$
\frac{S_1}{S_2} = \frac{\sum_{l=1}^{|\mathbf{i}_m^n|} \sum_{v=1; i \in (\mathbf{i}_1^m)_v}^{|\mathbf{i}_1^m|} a_i a_i}{\sum_{l=1; i \in (\mathbf{i}_1^n)_l}^{|\mathbf{i}_1^n|} a_i a_i} = \frac{|\mathbf{i}_m^n| \times |\mathbf{i}_1^m|}{|\mathbf{i}_1^n|} \tag{43}
$$

and when $r = 2$,

$$
\frac{S_1}{S_2} = \frac{\sum_{l=1}^{|\mathbf{i}_m^n|} \sum_{v=1; i,j \in (\mathbf{i}_2^m)_v}^{|\mathbf{i}_2^m|} a_i a_j}{\sum_{l=1; i \in (\mathbf{i}_2^n)_l}^{|\mathbf{i}_2^n|} a_i a_j} = \frac{|\mathbf{i}_m^n| \times |\mathbf{i}_2^m|}{|\mathbf{i}_2^n|}. \tag{44}
$$

Therefore, we compute Formula 42 by separately considering the diagonal and off-diagonal elements.

When $i = j$:

$$
\begin{aligned}
\sum_{r=1}^{k} \mathbb{E}_{\mathbf{Y}'}(\mathbf{1}_{m_i}^T \mathbf{K}'_{[r][r]} \mathbf{1}_{m_i})_{i=j} &= \frac{|\mathbf{i}_{m_1}^n| \times m_1}{|\mathbf{i}_{m_1}^n| \times n} \sum_{i=1}^{n} \mathbf{K}_{ii} + \frac{|\mathbf{i}_{m_1}^n| \times |\mathbf{i}_{m_2}^{n-m_1}| \times m_2}{|\mathbf{i}_{m_1}^n| \times |\mathbf{i}_{m_2}^{n-m_1}| \times n} \sum_{i=1}^{n} \mathbf{K}_{ii} + \cdots \\
&+ \frac{|\mathbf{i}_{m_1}^n| \times |\mathbf{i}_{m_2}^{n-m_1}| \times \cdots \times |\mathbf{i}_{m_k}^{n-m_1-\cdots-m_{k-1}}| \times m_k}{|\mathbf{i}_{m_1}^n| \times |\mathbf{i}_{m_2}^{n-m_1}| \times \cdots \times |\mathbf{i}_{m_k}^{n-m_1-\cdots-m_{k-1}}| \times n} \sum_{i=1}^{n} \mathbf{K}_{ii} \\
&= \frac{m_1}{n} \sum_{i=1}^{n} \mathbf{K}_{ii} + \frac{m_2}{n} \sum_{i=1}^{n} \mathbf{K}_{ii} + \cdots + \frac{m_k}{n} \sum_{i=1}^{n} \mathbf{K}_{ii} \\
&= \sum_{i=1}^{n} \mathbf{K}_{ii}
\end{aligned} \tag{45}
$$

When $i \neq j$:

$$\sum_{r=1}^{k} \mathbb{E}_{\boldsymbol{Y}'}(\mathbf{1}_{m_i}^T \mathbf{K}'_{[r][r]} \mathbf{1}_{m_i})_{i \neq j} = \frac{|\mathbf{i}_{m_1}^n| \times |\mathbf{i}_2^{m_1}|}{|\mathbf{i}_{m_1}^n| \times |\mathbf{i}_2^n|} \sum_{i,j;i \neq j}^{n} \mathbf{K}_{ij} + \frac{|\mathbf{i}_{m_1}^n| \times |\mathbf{i}_{m_2}^{n-m_1}| \times |\mathbf{i}_2^{m_2}|}{|\mathbf{i}_{m_1}^n| \times |\mathbf{i}_{m_2}^{n-m_1}| \times |\mathbf{i}_2^n|} \sum_{i,j;i \neq j}^{n} \mathbf{K}_{ij} + \cdots \tag{46}$$

$$+ \frac{|\mathbf{i}_{m_1}^n| \times |\mathbf{i}_{m_2}^{n-m_1}| \times \cdots \times |\mathbf{i}_{m_k}^{n-m_1-\cdots-m_{k-1}}| \times |\mathbf{i}_2^{m_k}|}{|\mathbf{i}_{m_1}^n| \times |\mathbf{i}_{m_2}^{n-m_1}| \times \cdots \times |\mathbf{i}_{m_k}^{n-m_1-\cdots-m_{k-1}}| \times |\mathbf{i}_2^n|} \sum_{i,j;i \neq j}^{n} \mathbf{K}_{ij}$$

$$= \frac{|\mathbf{i}_2^{m_1}|}{|\mathbf{i}_2^n|} \sum_{i,j;i \neq j}^{n} \mathbf{K}_{ij} + \frac{|\mathbf{i}_2^{m_2}|}{|\mathbf{i}_2^n|} \sum_{i,j;i \neq j}^{n} \mathbf{K}_{ij} + \cdots + \frac{|\mathbf{i}_2^{m_k}|}{|\mathbf{i}_2^n|} \sum_{i,j;i \neq j}^{n} \mathbf{K}_{ij}$$

$$= \sum_{r=1}^{k} \frac{|\mathbf{i}_2^{m_r}|}{|\mathbf{i}_2^n|} \sum_{i,j;i \neq j}^{n} \mathbf{K}_{ij}$$

Combining Formulas 45 and 46, we obtain:

$$\sum_{r=1}^{k} \mathbb{E}_{\boldsymbol{Y}'}(\mathbf{1}_{m_i}^T \mathbf{K}'_{[r][r]} \mathbf{1}_{m_i}) = \sum_{i=1}^{n} \mathbf{K}_{ii} + \sum_{r=1}^{k} \frac{|\mathbf{i}_2^{m_r}|}{|\mathbf{i}_2^n|} \sum_{i,j;i \neq j}^{n} \mathbf{K}_{ij} \tag{47}$$

So, we obtain an analytical solution of $\mathbb{E}_{\mathbf{Y}'}(d_{SED}(\mathbf{K}, \mathbf{Y}'))$:

$$\mathbb{E}_{\mathbf{Y}'}(d_{SED}(\mathbf{K}, \mathbf{Y}')) \tag{48}$$

$$= \|\mathbf{K}\|_F^2 + \sum_{i=1}^{k} m_i^2 - 2\left(\sum_{i=1}^{n} \mathbf{K}_{ii} + \sum_{r=1}^{k} \frac{|\mathbf{i}_2^{m_r}|}{|\mathbf{i}_2^n|} \sum_{i,j;i \neq j}^{n} \mathbf{K}_{ij}\right).$$

Based on Formula 39, 40, 41 and 48, the analytic solution of PSED is:

$$d_{PSED}(\mathbf{K}) = \tag{49}$$

$$2\left(\sum_{i=1}^{n} \mathbf{K}_{ii} + \sum_{r=1}^{k} \frac{|\mathbf{i}_2^{m_r}|}{|\mathbf{i}_2^n|} \sum_{i,j;i \neq j}^{n} \mathbf{K}_{ij} - \sum_{i=1}^{k} \mathbf{1}_{m_i} \mathbf{K}_{[i][i]} \mathbf{1}_{m_i}\right) \square$$

### 9.3. Proof of the computational efficiency of PSED

We provide the definition of PSED and the computational efficiency of analytical solutions.

For the definition of PSED, that is Formula 39, computational efficiency is divided into two parts, the first part is divided into three sub parts :(1) $\boldsymbol{Y} \times \boldsymbol{Y}^T$ : $\mathcal{O}(2kn^2)$; (2) $\mathbf{K} - \boldsymbol{Y}\boldsymbol{Y}^T$ : $\mathcal{O}(n^2)$; (3) $\|\mathbf{K} - \boldsymbol{Y}\boldsymbol{Y}^T\|_F^2$ : $\mathcal{O}(n^2)$. $\mathbb{E}_{\boldsymbol{Y}'}(d_{SED}(\mathbf{K}, \boldsymbol{Y}'))$ requires the calculation of all cases that follow the same distribution as the true label $Y$, involving a total of $\frac{n!}{m_1! m_2! \cdots m_k!}$ terms, so the computational efficiency of the second term is $\frac{n!}{m_1! m_2! \cdots m_k!} \times (\mathcal{O}(2kn^2) + \mathcal{O}(n^2) + \mathcal{O}(n^2))$. Therefore, the overall computational efficiency defined by PSED is $(\frac{n!}{m_1! m_2! \cdots m_k!} + 1) \times \mathcal{O}(2kn^2 + 2n^2)$.

For the analytic solution of PSED, computational efficiency is divided into three parts: (1) $\sum_{i=1}^{n} \mathbf{K}_{ii}$ : $\mathcal{O}(n)$; (2) $\sum_{r=1}^{k} \sum_{i,j;i \neq j}^{n} \mathbf{K}_{ij}$ : $\mathcal{O}(k(n^2 - n))$; (3) $\sum_{i=1}^{k} \mathbf{1}_{m_i} \mathbf{K}_{[i][i]} \mathbf{1}_{m_i}$ : $\mathcal{O}(\sum_{i=1}^{k} m_i^2)$. So, the overall computational efficiency of the analytical solution of PSED is $\mathcal{O}(kn^2 + (1-k)n + \sum_{i=1}^{k} m_i^2)$.

Due to the existence of inequalities:

$$(a + b + c)^2 \geq a^2 + b^2 + c^2 \tag{50}$$

Therefore, $\mathcal{O}(kn^2 + (1-k)n + \sum_{i=1}^{k} m_i^2) < \mathcal{O}(2kn^2 + 2n^2)$ and $(\frac{n!}{m_1! m_2! \cdots m_k!} + 1)$ is large, the computational efficiency of the analytical solution of PSED is significantly higher than that of the PSED definition. In other words, the analytical solution of PSED has more effective computational efficiency.

## 9.4. Proof of properties 3 and 4

**Property 3** (**Bias of** $d_{SED}$) For the matrices $\boldsymbol{I}$ and $\boldsymbol{J}$, when $\sum_{i=1}^{k} m_i^2 > \frac{n^2+n}{2}$, we have $d_{SED}(\boldsymbol{I}) > d_{SED}(\boldsymbol{J})$; otherwise, $d_{SED}(\boldsymbol{I}) < d_{SED}(\boldsymbol{J})$. **Proof** According to the analytic solution of $d_{SED}$:

$$d_{SED}(\mathbf{K}, \boldsymbol{Y}) = \|\mathbf{K} - \boldsymbol{Y}\boldsymbol{Y}^T\|_F^2 \tag{51}$$

$$= \|\mathbf{K}\|_F^2 + \sum_{i=1}^{k} m_i^2 - 2\sum_{i=1}^{k} \mathbf{1}_{m_i}^T \mathbf{K}_{[i][i]} \mathbf{1}_{m_i}$$

When $\mathbf{K} = \boldsymbol{J}$, we have:

$$d_{SED}(\boldsymbol{J}) \tag{52}$$

$$= \|\boldsymbol{J}\|_F^2 - 2\sum_{i=1}^{k} \mathbf{1}_{m_i}^T \boldsymbol{J}_{m_i \times m_i} \mathbf{1}_{m_i} + \sum_{i=1}^{k} m_i^2$$

$$= n^2 - \sum_{i=1}^{k} m_i^2.$$

When $\mathbf{K} = \boldsymbol{I}$, we have:

$$d_{SED}(\boldsymbol{I}) \tag{53}$$

$$= \|\boldsymbol{I}\|_F^2 - 2\sum_{i=1}^{k} \mathbf{1}_{m_i}^T \boldsymbol{I}_{m_i \times m_i} \mathbf{1}_{m_i} + \sum_{i=1}^{k} m_i^2$$

$$= \sum_{i=1}^{k} m_i^2 - n.$$

Then, we have:

$$d_{SED}(\boldsymbol{I}) - d_{SED}(\boldsymbol{J}) \tag{54}$$

$$= \sum_{i=1}^{k} m_i^2 - n - (n^2 - \sum_{i=1}^{k} m_i^2)$$

$$= 2\sum_{i=1}^{k} m_i^2 - n - n^2.$$

Thus, we obtain the conclusion. $\square$

**Property 4** (**Unbiased of** $d_{PSED}$) For any matrix $\boldsymbol{A}$ in $\boldsymbol{N}_a = \{(1-a)\boldsymbol{I} + a\boldsymbol{J}, 0 \le a \le 1\}$, where $\boldsymbol{I}$ is the identity matrix and $\boldsymbol{J}$ is the full one matrix. We have $d_{PSED}(\boldsymbol{A}) = 0$.

**Proof** In fact, $\boldsymbol{A}$ is the matrix with the diagonal is 1 and the other elements are $a$. For $\boldsymbol{Y}$ and $\boldsymbol{Y}'$, their sub block matrices are the same, that is: $\boldsymbol{A}_{[i][i]} = \boldsymbol{A}'_{[i][i]} = ((1-a)\boldsymbol{I} + a\boldsymbol{J})_{m_i \times m_i}$. Combining with the analytic solution of $d_{SED}$, we have that $d_{SED}(\boldsymbol{A}, \boldsymbol{Y}) = d_{SED}(\boldsymbol{A}, \boldsymbol{Y}') = \mathbb{E}_{\boldsymbol{Y}'}(d_{SED}(\boldsymbol{A}, \boldsymbol{Y}'))$. Thus, we have $d_{PSED}(\boldsymbol{A}) = 0$

## 9.5. Proof of Theorem 5.2

*Proof.* To use the concentration bound in Theorem 5.2, we firstly need to investigate the change in the loss when a single object is modified. Secondly, we need give the exponential Orlicz norm bound of the loss change. The definitions and properties of the sub Gaussian norm used in the proof are provided at the end of this section.

**For the first step**, without loss of generality, we assume that the changed sample belongs to the first category. In this case,

we have,

$$|\hat{d}_{PSED}(\hat{\mathbf{K}}(\mathcal{S}_n)) - \hat{d}_{PSED}(\hat{\mathbf{K}}(\mathcal{S}_{n,k}))| \tag{55}$$

$$= \left|2(\hat{\mathbf{K}}_{kk} - \hat{\mathbf{K}}_{k'k'}) + 4C\left(\sum_{j:j\neq k}\hat{\mathbf{K}}_{kj} - \sum_{j:j\neq k}\hat{\mathbf{K}}_{k'j}\right) - 4(\hat{\mathbf{K}}_{k[1]}\mathbf{1}_{m_1} - \hat{\mathbf{K}}_{k'[1]}\mathbf{1}_{m_1})\right| \tag{56}$$

$$\leq \left|2(\hat{\mathbf{K}}_{kk} - \hat{\mathbf{K}}_{k'k'})\right| + 4C\left|\left(\sum_{j:j\neq k}\hat{\mathbf{K}}_{kj} - \sum_{j:j\neq k}\hat{\mathbf{K}}_{k'j}\right)\right| + 4\left|(\hat{\mathbf{K}}_{k[1]}\mathbf{1}_{m_1} - \hat{\mathbf{K}}_{k'[1]}\mathbf{1}_{m_1})\right| \tag{57}$$

where $C = \sum_{r=1}^{k}|\mathbf{i}_2^{m_r}|/|\mathbf{i}_2^n|$.

From Lemma 13 in (Greenfeld & Shalit, 2020), we know that assume $K(z,y) = \exp\left(-\gamma(z-y)^2\right)$, as is the case with RBF kernels, and suppose $\|\boldsymbol{x}\| \leq \frac{M}{2}$ for all $\boldsymbol{x} \in y$. Then $K(\cdot,\cdot)$ is $\gamma M$-Lipschitz for all $\boldsymbol{x} \in \boldsymbol{X}$. Therefore, we have,

$$|\hat{\mathbf{K}}_{kk} - \hat{\mathbf{K}}_{k'k'}| \tag{58}$$
$$= |K(h(\boldsymbol{x}_k), h(\boldsymbol{x}_k)) - K(h(\boldsymbol{x}_k), h(\boldsymbol{x}'_k)) + K(h(\boldsymbol{x}_k), h(\boldsymbol{x}'_k)) - K(h(\boldsymbol{x}'_k), h(\boldsymbol{x}'_k))| \tag{59}$$
$$\leq 2|K(h(\boldsymbol{x}_k), h(\boldsymbol{x}_k)) - K(h(\boldsymbol{x}_k), h(\boldsymbol{x}'_k))| \tag{60}$$
$$\leq 2\gamma M\|h(\boldsymbol{x}_k) - h(\boldsymbol{x}'_k))\|_2 \leq 2\gamma M\|h(\boldsymbol{x}_k) - h(\boldsymbol{x}'_k))\|_1 \tag{61}$$

$$\leq 2\gamma M\|\boldsymbol{x}_k - \boldsymbol{x}'_k\|_1 \prod_{l=1}^{L-1}\rho_l\|\boldsymbol{W}_l\|_1 \tag{62}$$

By a combination of Eq. (58) and the triangle inequality, for the second term and third term of Eq. (55), respectively, we have:

$$\left|\left(\sum_{j:j\neq k}\hat{\mathbf{K}}_{kj} - \sum_{j:j\neq k}\hat{\mathbf{K}}_{k'j}\right)\right| \leq 2\gamma M(n-1)\|\boldsymbol{x}_k - \boldsymbol{x}'_k\|_1 \prod_{l=1}^{L-1}\rho_l\|\boldsymbol{W}_l\|_1, \tag{63}$$

$$\left|(\hat{\mathbf{K}}_{k[1]}\mathbf{1}_{m_1} - \hat{\mathbf{K}}_{k'[1]}\mathbf{1}_{m_1})\right| \leq 2\gamma M m_1\|\boldsymbol{x}_k - \boldsymbol{x}'_k\|_1 \prod_{l=1}^{L-1}\rho_l\|\boldsymbol{W}_l\|_1. \tag{64}$$

**For the second step**, sequentially by Property 6 and 7, we have,

$$\left\|\|\boldsymbol{x}_k - \boldsymbol{x}'_k\|_1 \prod_{l=1}^{L-1}\rho_l\|\boldsymbol{W}_l\|_1\right\|_{\psi_1} \tag{65}$$

$$\leq \left\|\|\boldsymbol{x}_k - \boldsymbol{x}'_k\|_1\right\|_{\psi_L} \prod_{l=1}^{L-1}\rho_l\left\|\|\boldsymbol{W}_l\|_1\right\|_{\psi_L} \tag{66}$$

$$\leq \sqrt{d}\|\boldsymbol{x}_k - \boldsymbol{x}'_k\|_{\psi_L} \prod_{l=1}^{L-1}\rho_l\sqrt{d_l}\|\boldsymbol{W}_l\|_{\psi_L} \tag{67}$$

$$\leq \sqrt{d}\|diam(\boldsymbol{x})\|_{\psi_L} \prod_{l=1}^{L-1}\rho_l\sqrt{d_l}\|\boldsymbol{W}_l\|_{\psi_L}, \tag{68}$$

where the last inequality is according to the definition of $\psi_q$ norm. $\square$

Above all, we have,

$$|\hat{d}_{PSED}(\hat{\mathbf{K}}(\mathcal{S}_n)) - \hat{d}_{PSED}(\hat{\mathbf{K}}(\mathcal{S}_{n,k}))| \tag{69}$$

$$\leq (2 + 4C(n-1) + 4m_1)2\gamma M\left\|\|\boldsymbol{x}_k - \boldsymbol{x}'_k\|_1 \prod_{l=1}^{L-1}\rho_l\|\boldsymbol{W}_l\|_1\right\|_{\psi_1} \tag{70}$$

$$\leq (2 + 4C(n-1) + 4m_1)2\gamma M\sqrt{d}\|diam(\boldsymbol{x})\|_{\psi_L} \prod_{l=1}^{L-1}\rho_l\sqrt{d_l}\|\boldsymbol{W}_l\|_{\psi_L}. \tag{71}$$

Thus, we obtain the final result.

### 9.5.1. PROOFS OF PROPERTIES

A random variable with a finite $\|X\|_{\psi_q}$ admits a tail satisfying (Vershynin, 2018),

$$\mathbb{P}\left(|x| \geq t\right) \leq 2 \exp\left(-\frac{t^q}{\|X\|_{\psi_q}^q}\right). \tag{72}$$

From this tail, we can observe that the smaller the $\|X\|_{\psi_q}$ norm, the more concentrated the distribution of variables.

**Theorem 9.2.** *(**Young's Inequality**) Let $a_1, ..., a_L \geq 0, p_1, ..., p_L > 1, \sum_{i=1}^{L} \frac{1}{p_i} = 1$, there are Young's Inequality,*

$$\prod_{i=1}^{L} a_i \leq \sum_{i=1}^{L} \frac{a_i^{p_i}}{p_i},$$

*the equality holds when $a_1^{p_1} = ... = a_i^{p_i} = ... = a_L^{p_L}$.*

**Proof of Property 5**

*Proof.* Suppose $\|X\|_{\psi_2} = c_1/2$ and $\|Y\|_{\psi_2} = c_2/2$, then by definition,

$$\mathbb{E}\left[\exp\left|\frac{2X}{c_1}\right|\right] \leq 2, \quad \mathbb{E}\left[\exp\left|\frac{2Y}{c_2}\right|\right] \leq 2. \tag{73}$$

We have,

$$\mathbb{E}\exp\left(\left|\frac{X+Y}{c_1+c_2}\right|\right) \tag{74}$$

$$\leq \mathbb{E}\exp\left(\left|\frac{X}{c_1+c_2}\right| + \left|\frac{Y}{c_1+c_2}\right|\right) \tag{75}$$

$$\leq \mathbb{E}\exp\left(\left|\frac{X}{c_1}\right| + \left|\frac{Y}{c_2}\right|\right) \tag{76}$$

$$\leq \mathbb{E}\left[\exp\left|\frac{X}{c_1}\right| \exp\left|\frac{Y}{c_2}\right|\right] \tag{77}$$

$$\leq \frac{1}{2}\mathbb{E}\left[\exp\left|\frac{2X}{c_1}\right| + \exp\left|\frac{2Y}{c_2}\right|\right] \leq 2, \tag{78}$$

where the first inequality is based on the triangle inequality and the last inequality is based on the Young's inequality. $\square$

**Proof of Property 6**

*Proof.* We assume that $\|X_i\|_{\psi_L} = c_i$, then,

$$\mathbb{E}\left[\exp\left|\frac{X_i}{c_i}\right|^L\right] \leq 2.$$

There exists that,

$$\mathbb{E}\exp\left(\prod_{i=1}^{L}\left|\frac{X_i}{c_i}\right|\right) \le \mathbb{E}\exp\left(\sum_{i=1}^{L}\frac{\left|\frac{X_i}{c_i}\right|^L}{L}\right)$$

$$= \mathbb{E}\left[\prod_{i=1}^{L}\exp\left(\frac{\left|\frac{X_i}{c_i}\right|^L}{L}\right)\right]$$

$$\le \frac{1}{L}\mathbb{E}\left[\sum_{i=1}^{L}\left|\frac{X_i}{c_i}\right|^L\right]$$

$$\le 2,$$

where the first and the second inequalities are based on Young's inequality. $\qquad\square$

**Proof of Property 7**

*Proof.* There exists,

$$\mathbb{E}\left[\exp\left|\frac{\|\boldsymbol{X}\|_1}{c}\right|^q\right] = \mathbb{E}\left[\exp\left|\frac{\sum_{i=1}^{d}|\boldsymbol{X}^i|}{c}\right|^q\right] \tag{79}$$

$$= \mathbb{E}\left[\exp\left|\frac{\left\langle\boldsymbol{X},\frac{1}{\sqrt{d}}\mathbf{1}_{d\times1}\right\rangle}{\frac{c}{\sqrt{d}}}\right|^q\right], \tag{80}$$

where $\mathbf{1}_{d\times1}$ is a column vector of which elements are all 1. We assume that $\|\|\boldsymbol{X}\|_1\|_{\psi_q} = c$. Then by definition, we obtain,

$$\left\|\left\langle\boldsymbol{X},\frac{1}{\sqrt{d}}\mathbf{1}_{d\times1}\right\rangle\right\|_{\psi_q} = \frac{c}{\sqrt{d}}. \tag{81}$$

Because $\frac{1}{\sqrt{d}}\mathbf{1}_{d\times1}\in\mathbb{S}^{d-1}$, we have that

$$\frac{c}{\sqrt{d}} \le \sup_{v\in\mathbb{S}^{d-1}}\|\langle\boldsymbol{X},v\rangle\|_{\psi_q}. \tag{82}$$

$\qquad\square$

## 10. The algorithm process diagram of the method

The specific algorithm process of the method used in this paper is shown in Algorithm 1, where $RL$ is the representation layer and $CL$ is the classification layer.

---

**Algorithm 1** Model Construction

---

**INPUT**: The training sample features and labels of features: $\mathbf{X_{train}}, Y_{train}$.
The maximum number of iterations *epo*.
**OUTPUT**: The model parameters $\theta_{RL}, \theta_{CL}$.

 1: Initialize model parameters and learning rate: $\boldsymbol{W}_{RL}, \boldsymbol{W}_{CL}, \eta$.
 2: **for** epoch=1:*epo* **do**
 3:     $(\mathbf{Z_{train}}) \leftarrow RL(\mathbf{X_{train}};\boldsymbol{W}_{RL})$.
 4:     Calculate the loss of PSED and perform back propagation and update parameters $\boldsymbol{W}_{RL}$.
 5:     $(\tilde{Y}) \leftarrow CL(\mathbf{Z_{train}};\boldsymbol{W}_{CL})$.
 6:     Calculate the loss of CE and perform back propagation and update parameters $\boldsymbol{W}_{CL}$.
 7: **end for**

---

# 11. Experiment

### 11.0.1. THE DATASETS AND MODEL STRUCTURE PARAMETERS

We provide a detailed description of the dataset and download links. For more detailed information, please refer to Tables 4 and 5. Among them, the imbalance ratio refers to the ratio between the most common and rare categories in the dataset. The description of the image dataset is as follows:

- **MPEG Dataset**: The Moving Picture Experts Group (MPEG) dataset includes video sequences designed for testing video encoding and transmission algorithms. The dataset consists of videos captured under various scenarios and conditions, making it ideal for evaluating the performance of video encoding techniques. We show some images of the dataset as shown in Figure 8 (a).

- **MNIST Dataset**: The Modified National Institute of Standards and Technology (MNIST) dataset is widely used for digit recognition tasks. It contains 60,000 training samples and 10,000 testing samples, each represented by a 28x28 grayscale image of digits from 0 to 9.

- **Pendigits Dataset**: The Pendigits dataset is dedicated to handwritten digit recognition. It includes over 10,000 32x32 grayscale images of handwritten digits (0-9), with separate training and testing sets.

- **Caltech-101 Dataset**: A widely used object recognition dataset, Caltech-101 contains approximately 9,000 images across 101 object categories, including animals, vehicles, food, and furniture. The dataset features images taken in diverse real-world settings, offering a robust benchmark for image classification tasks. We show some images of the dataset as shown in Figure 8 (b).

- **ImageNet Dataset**: ImageNet is a comprehensive image recognition database with over 14 million labeled images spanning more than 20,000 distinct classes. It is extensively used for evaluating the performance of image classification models. We show some images of the dataset as shown in Figure 8 (c).

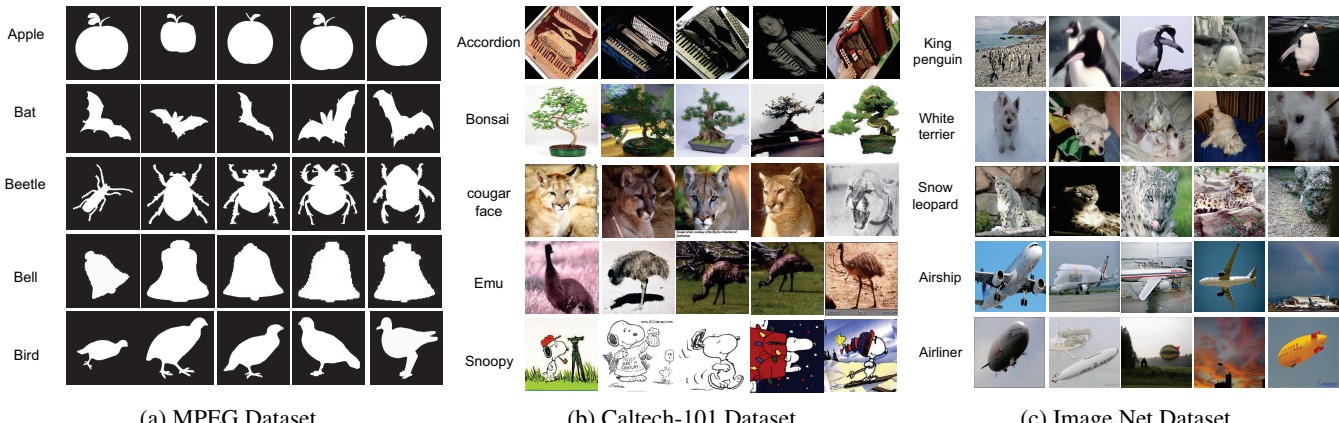

(a) MPEG Dataset        (b) Caltech-101 Dataset        (c) Image Net Dataset

*Figure 8.* Example pictures of image datasets.

To ensure consistency in the evaluation, each dataset is randomly divided into training, validation, and testing sets in a 5:2:3 ratio. This hierarchical approach ensures that the performance evaluation of the model at different stages of training is not affected by randomness. To investigate the impact of model architecture on performance, we evaluated configurations with 3, 5, and 8 layers. This change allows us to compare in detail how model complexity affects performance under different loss functions. The training process uses the Adam optimizer, which is widely favored for its efficiency in adjusting learning rates. The learning rate of each dataset has been fine tuned to achieve optimal convergence performance. Training for up to 60 epochs provides ample time for model learning and reduces the risk of overfitting. We also customized hidden layers and regularization parameters for each dataset. This customization takes into account the unique characteristics and complexity of each dataset, ensuring that the model architecture is best suited for optimal performance. The batch size is set to 16. For feature extraction, ImageNet employs the self-supervised learning model MOCO V3 (Chen et al., 2021), while all other datasets utilize VGG (Fernandez-Delgado et al., 2014).

*Table 4.* Description of the Datasets

| ID | Name | Object | Dimension | Class | Imbalanced Ratio |
|----|------|--------|-----------|-------|------------------|
| 1 | Wine Quality | 734 | 11 | 2 | 12.85:1 |
| 2 | Blood Transfusion Service Center | 748 | 4 | 2 | 3.20:1 |
| 3 | Energy Efficiency | 768 | 9 | 2 | 1.87:1 |
| 4 | Tic-Tac-Toe Endgame | 957 | 9 | 2 | 1.88:1 |
| 5 | Oocytes-Merluccius-Nucleus-4d | 1022 | 41 | 2 | 2.03:1 |
| 6 | QSAR Biodegradation | 1055 | 41 | 2 | 1.96:1 |
| 7 | Yeast | 1484 | 8 | 10 | 92.60:1 |
| 8 | Semeion Handwritten Digit | 1593 | 156 | 10 | 1.05:1 |
| 9 | Steel Plates Faults | 1941 | 27 | 7 | 12.24:1 |
| 10 | Cardiotocography | 1950 | 21 | 2 | 5.61:1 |
| 11 | Ozone Level Detection | 2536 | 72 | 2 | 33.74:1 |
| 12 | SkillCraft1 Master Table | 3343 | 21 | 2 | 1.03:1 |
| 13 | Gender Gap in Spanish WP | 3355 | 21 | 2 | 17.95:1 |
| 14 | Waveform Database Generator | 3679 | 21 | 2 | 18.26:1 |
| 15 | Abalone | 4177 | 8 | 2 | 2.16:1 |
| 16 | Page Blocks Classification | 5242 | 10 | 2 | 14.93:1 |
| 17 | Ringnorm | 7400 | 20 | 2 | 1.02:1 |
| 18 | Mushroom | 8124 | 21 | 2 | 1.07:1 |
| 19 | Nursery | 12960 | 8 | 4 | 13.09:1 |
| 20 | Adult | 32561 | 14 | 2 | 3.15:1 |
| 21 | Mpeg | 1400 | 6000 | 70 | 1.00:1 |
| 22 | Mnist | 6996 | 784 | 10 | 1.25:1 |
| 23 | Pendigits | 7494 | 16 | 10 | 1.08:1 |
| 24 | Caltech-101 | 8641 | 256 | 101 | 19.46:1 |
| 25 | ImageNet | 13000 | 256 | 10 | 1.00:1 |

*Table 5.* Addresses of Datasets

| ID | Data Address |
|----|--------------|
| 1 | https://archive.ics.uci.edu/dataset/186/wine+quality |
| 2 | https://archive.ics.uci.edu/dataset/176/blood+transfusion+service+center |
| 3 | https://archive.ics.uci.edu/dataset/242/energy+efficiency |
| 4 | https://archive.ics.uci.edu/dataset/101/tic+tac+toe+endgame |
| 5 | https://gitlab.citius.gal/jorge.suarez/fishovary/-/tree/4e434ce0c6fa93b7d2afe67a4c941a178613fa85 |
| 6 | https://archive.ics.uci.edu/dataset/254/qsar+biodegradation |
| 7 | https://archive.ics.uci.edu/dataset/110/yeast |
| 8 | https://archive.ics.uci.edu/dataset/178/semeion+handwritten+digit |
| 9 | https://archive.ics.uci.edu/dataset/198/steel+plates+faults |
| 10 | https://archive.ics.uci.edu/dataset/193/cardiotocography |
| 11 | https://archive.ics.uci.edu/dataset/172/ozone+level+detection |
| 12 | https://archive.ics.uci.edu/dataset/272/skillcraft1+master+table+dataset |
| 13 | https://archive.ics.uci.edu/dataset/852/gender+gap+in+spanish+wp |
| 14 | https://archive.ics.uci.edu/dataset/107/waveform+database+generator+version+1 |
| 15 | https://archive.ics.uci.edu/dataset/1/abalone |
| 16 | https://archive.ics.uci.edu/dataset/78/page+blocks+classification |
| 17 | https://www.cs.toronto.edu/ delve/data/ringnorm/desc.html |
| 18 | https://archive.ics.uci.edu/dataset/73/mushroom |
| 19 | https://archive.ics.uci.edu/dataset/76/nursery |
| 20 | https://archive.ics.uci.edu/dataset/2/adult |
| 21 | https://dabi.temple.edu/external/shape/MPEG7/dataset.html |
| 22 | https://tensorflow.google.cn/datasets/catalog/mnist |
| 23 | https://www.dbs.ifi.lmu.de/research/outlier-evaluation/DAMI/literature/PenDigits/ |
| 24 | https://tensorflow.google.cn/datasets/catalog/caltech101 |
| 25 | https://paperswithcode.com/sota/image-clustering-on-imagenet-10 |

### 11.0.2. THE EVALUATING MEASURE

In this analysis, we evaluate the model using two key performance metrics: accuracy and F-measure, which are defined as follows:

$$\text{Accuracy} = \frac{1}{n}\sum_{i=1}^{n}\mathbb{I}(Y_i = \hat{Y}_i), \tag{83}$$

$$\text{F-measure} = \frac{2\text{Precision} \times \text{Recall}}{\text{Precision} + \text{Recall}}, \tag{84}$$

where

$$\text{Precision} = \frac{\sum_{i=1}^{n}\sum_{j=1}^{k}\mathbb{I}(\hat{Y}_{ij} = 1, Y_{ij} = 1)}{\sum_{i=1}^{n}\sum_{j=1}^{k}\mathbb{I}(\hat{Y}_{ij} = 1)}, \tag{85}$$

$$\text{Recall} = \frac{\sum_{i=1}^{n}\sum_{j=1}^{k}\mathbb{I}(\hat{Y}_{ij} = 1, Y_{ij} = 1)}{\sum_{i=1}^{n}\sum_{j=1}^{k}\mathbb{I}(Y_{ij} = 1)}, \tag{86}$$

$\hat{Y}_{ij}$ is the predicted label for sample $i$ and class $j$, $Y_{ij}$ is the true label for sample $i$ and class $j$, and $C$ is the number of class. Additionally, $\mathbb{I}$ is the indicator function, which takes a value of 1 when the condition inside the parentheses is true, and 0 otherwise.

## 11.1. Experimental

### 11.1.1. COMPARISON OF CLASSIFICATION PERFORMANCE

To assess whether there are statistically significant differences between the proposed method and other approaches, we first conduct a one-sided t-test. The null hypothesis $H_0$ assumes the proposed method is inferior to other methods, while the alternative hypothesis $H_1$ posits that the proposed method outperforms the others. The significance level for the test is set to 0.05. If the p-value is below this threshold, $H_0$ is rejected, indicating that the proposed method demonstrates statistically significant superiority. As shown in Tables 6 and 7, if our method outperforms others significantly, a black dot will be added next to the method. As shown in the tables, the PSED-based loss function demonstrates superior performance in terms of average convergence accuracy and F-measure, with values surpassing other methods on most datasets.

### 11.1.2. SIGNIFICANCE TEST

To investigate whether the proposed algorithm exhibits significant performance differences compared to baseline methods, we apply the Friedman test. This non-parametric test evaluates the rankings of multiple related samples across multiple datasets, enabling us to determine whether significant differences exist among the four methods being compared. The null hypothesis ($H_0$) assumes no significant differences among the methods, while the alternative hypothesis ($H_1$) suggests that at least two of the methods differ significantly. A p-value threshold of 0.05 is used in this analysis. If the p-value is below this threshold, $H_0$ is rejected, indicating that significant differences exist among at least two of the algorithms. Upon obtaining a significant result from the Friedman test, we perform the Nemenyi post-hoc test to identify which specific pairs of algorithms differ significantly. The Nemenyi test calculates the critical difference (CD) value, which is subsequently visualized in a CD diagram. This diagram offers a clear representation of the average ranks of each algorithm, with horizontal bars denoting significant differences between algorithm pairs.

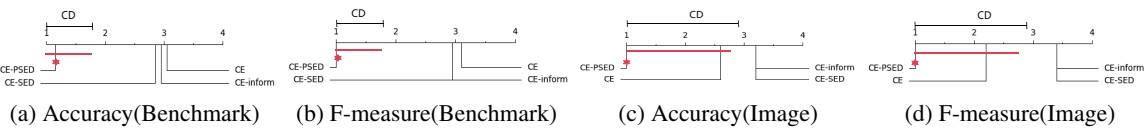

(a) Accuracy(Benchmark)      (b) F-measure(Benchmark)      (c) Accuracy(Image)      (d) F-measure(Image)

*Figure 9.* CD diagrams w.r.t. Accuracy and F-measure when model layer is 5.

*Table 6.* Accuracy and F-measure based on different loss functions when model layers is 5.

| Data | Accuracy | | | | F-measure | | | |
|---|---|---|---|---|---|---|---|---|
| | CE | CE-SED | CE-inform | CE-PSED | CE | CE-SED | CE-inform | CE-PSED |
| 1 | 0.9222±0.0009 | 0.9267±0.0000 | 0.8421±0.0731 | **0.9276±0.0000** | 0.8907±0.0002 | 0.8923±0.0000 | 0.8045±0.0779 | **0.8928±0.0000** |
| 2 | 0.6929±0.0297● | 0.7076±0.0267● | 0.7591±0.0003● | **0.7769±0.0003** | 0.5950±0.0316● | 0.6027±0.0318 | 0.6605±0.0000 ● | **0.7124±0.0021** |
| 3 | 0.6506±0.0000 ● | 0.6489±0.0000 ● | 0.6182±0.0094● | **0.7667±0.0005** | 0.5168±0.0003● | 0.5126±0.0000 ● | 0.4932±0.0151 ● | **0.7631±0.0005** |
| 4 | 0.6038±0.0128● | 0.6531±0.0000 ● | 0.6451±0.0012● | **0.7920±0.0001** | 0.4830±0.0132 ● | 0.5164±0.0000● | 0.5208±0.0001 ● | **0.7820±0.0001** |
| 5 | 0.6616±0.0002 | 0.6694±0.0000 | 0.6713±0.0000 | **0.7837±0.0004** | 0.5416±0.0006 | 0.5447±0.0005 | 0.5455±0.0005 | **0.7732±0.0003** |
| 6 | 0.6741±0.0002● | 0.6334±0.0109● | 0.6625±0.0000● | **0.8770±0.0003** | 0.5611±0.0029● | 0.5013±0.0144 ● | 0.5291±0.0001● | **0.8769±0.0003** |
| 7 | 0.2830±0.0040 ● | 0.2910±0.0020 ● | 0.2944±0.0008 ● | **0.5231±0.0014** | 0.1539±0.0030● | 0.1734±0.0018● | 0.1620±0.0008● | **0.4951±0.0016** |
| 8 | 0.2372±0.0041 ● | 0.2701±0.0016 ● | 0.2969±0.0040● | **0.8749±0.0003** | 0.1605±0.0051● | 0.1771±0.0011● | 0.2100±0.0048● | **0.8747±0.0004** |
| 9 | 0.4153±0.0023 ● | 0.4348±0.0046 ● | 0.4544±0.0028● | **0.6487±0.0002** | 0.2891±0.0031● | 0.3183±0.0046● | 0.3355±0.0042● | **0.6185±0.0004** |
| 10 | 0.8489±0.0000● | 0.8489±0.0000● | 0.8489±0.0000● | **0.9246±0.0002** | 0.7795±0.0000● | 0.7795±0.0000● | 0.7795±0.0000● | **0.9237±0.0002** |
| 11 | 0.9711±0.0000 | 0.9711±0.0000 | 0.9711±0.0000 | **0.9699±0.0000** | 0.9568±0.0000 | 0.9568±0.0000 | 0.9568±0.0000 | **0.9616±0.0000** |
| 12 | 0.9480±0.0000 | 0.9512±0.0001 | 0.9484±0.0000 | **0.9513±0.0000** | 0.9480±0.0000 | 0.9512±0.0001 | 0.9484±0.0000 | **0.9513±0.0000** |
| 13 | 0.9474±0.0000● | 0.9474±0.0000● | 0.9474±0.0000● | **0.9809±0.0000** | 0.9218±0.0000 ● | 0.9218±0.0000 ● | 0.9218±0.0000 ● | **0.9807±0.0000** |
| 14 | 0.9484±0.0000● | 0.9484±0.0000● | 0.9484±0.0000● | **0.9862±0.0000** | 0.9232±0.0000● | 0.9232±0.0000 ● | 0.9232±0.0000 ● | **0.9862±0.0000** |
| 15 | 0.5803±0.0006● | 0.5897±0.0006● | 0.5889±0.0002● | **0.6930±0.0001** | 0.5537±0.0053● | 0.5815±0.0015 ● | 0.5821±0.0006 ● | **0.6931±0.0001** |
| 16 | 0.9519±0.0003● | 0.9445±0.0002● | 0.9479±0.0002 ● | **0.9870±0.0000** | 0.9334±0.0010● | 0.9200±0.0007● | 0.9266±0.0008● | **0.9869±0.0000** |
| 17 | 0.9479±0.0001● | 0.9456±0.0001 ● | 0.9459±0.0002● | **0.9726±0.0000** | 0.9479±0.0001● | 0.9456±0.0001● | 0.9459±0.0002● | **0.9726±0.0000** |
| 18 | 0.9884±0.0000● | 0.9885±0.0000 ● | 0.9872±0.0000● | **1.0000±0.0000** | 0.9884±0.0000● | 0.9885±0.0000● | 0.9872±0.0000● | **1.0000±0.0000** |
| 19 | 0.9049±0.0000● | 0.9061±0.0000● | 0.9064±0.0000● | **0.9810±0.0000** | 0.8942±0.0000● | 0.8951±0.0000● | 0.8956±0.0000 ● | **0.9809±0.0000** |
| 20 | 0.8442±0.0000● | 0.8445±0.0000 | 0.8451±0.0000 | **0.8477±0.0000** | 0.8389±0.0000● | 0.8391±0.0000 | 0.8399±0.0000 | **0.8417±0.0000** |
| 21 | 0.0721±0.0002● | 0.0552±0.0003 ● | 0.0624±0.0006● | **0.6283±0.0001** | 0.0344±0.0001● | 0.0268±0.0001● | 0.0294±0.0003● | **0.6148±0.0002** |
| 22 | 0.7236±0.0031 ● | 0.7255±0.0023● | 0.7222±0.0030● | **0.8963±0.0000** | 0.7099±0.0046● | 0.7044±0.0029● | 0.7033±0.0043● | **0.8962±0.0000** |
| 23 | 0.6603±0.0053● | 0.6281±0.0059● | 0.6374±0.0052● | **0.9777±0.0000** | 0.6441±0.0067● | 0.6023±0.0090● | 0.6120±0.0078● | **0.9777±0.0000** |
| 24 | 0.2413±0.0001● | 0.2359±0.0000 ● | 0.2421±0.0000● | **0.4453±0.0001** | 0.1324±0.0001● | 0.1279±0.0001● | 0.1307±0.0001 ● | **0.4107±0.0001** |
| 25 | 0.9713±0.0000 ● | 0.9727±0.0000● | 0.9712±0.0000● | **0.9770±0.0000** | 0.9714±0.0000● | 0.9728±0.0000● | 0.9713±0.0000 ● | **0.9770±0.0000** |

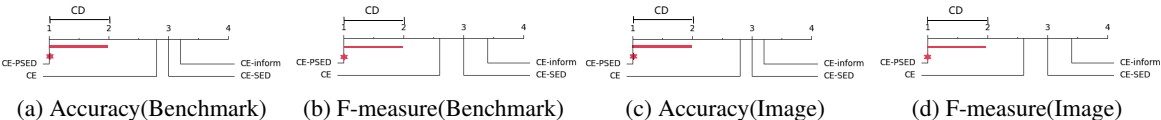

(a) Accuracy(Benchmark)  (b) F-measure(Benchmark)  (c) Accuracy(Image)  (d) F-measure(Image)

*Figure 10.* CD diagrams w.r.t. Accuracy and F-measure when model layer is 8.

### 11.1.3. CONVERGENCE ANALYSIS

The Figure 11 to 13 show the performance of the four methods over the training epoch in all benchmark datasets and image datasets, where the points on each line represent the average accuracy of the corresponding period. The results show that the CE-PSED method exhibits significant performance advantages in most datasets and training epochs, and can quickly converge, which fully demonstrates its robustness and effectiveness on different datasets.

### 11.1.4. NETWORK LAYER ANALYSIS

To verify the effectiveness of the CE-PSED method at different network layers, this study set the model layers to 5 and 8, respectively. Tables 6 and 7 show the accuracy and F-measure values of four methods at different levels. The results show that as the number of network layers increased, the F-measure values of other methods significantly decreased, while the F-measure value of the CE-PSED method decrease less and remain the highest. This indicates that other models experience feature representation collapse as the number of layers increases, that is, the features tend to be the same, while the CE-PSED method performs well at different layers, effectively avoiding feature collapse and ensuring that the model maintains good feature learning and classification capabilities in deep structures.

### 11.1.5. ANALYSIS OF DISCERNMENT ABILITY

To insight the discernment ability of the four methods, we analyze the feature representations of the last hidden layer of the model. Table 8 to Table 10 present the Euclidean distances and information entropies between the similarity matrices and $\mathbf{Y}\mathbf{Y}^T$ matrix for all dataset when model layers are 3, 5,and 8, where each row is a dataset, and the columns are divided

*Table 7.* Accuracy and F-measure based on different loss functions when model layers is 8.

| Data | Accuracy | | | | F-measure | | | |
|---|---|---|---|---|---|---|---|---|
| | CE | CE-SED | CE-inform | CE-PSED | CE | CE-SED | CE-inform | CE-PSED |
| 1 | 0.9249±0.0012 | 0.8421±0.0731 | 0.9267±0.0001 | **0.9276±0.0000** | 0.8921±0.0003 | 0.8045±0.0780 | 0.8930±0.0000 | **0.8928±0.0000** |
| 2 | 0.6871±0.0345 | 0.7538±0.0004 | 0.6827±0.0415● | **0.7689±0.0002** | 0.5953±0.0342 | 0.6557±0.0000 | 0.5886±0.0439 ● | **0.6798±0.0011** |
| 3 | 0.6455±0.0000 ● | 0.6147±0.0089● | 0.5892±0.0159● | **0.7580±0.0007** | 0.5119±0.0005● | 0.4804±0.0114● | 0.4452±0.0194 ● | **0.7575±0.0006** |
| 4 | 0.6240±0.0097● | 0.6184±0.0092● | 0.6528±0.0000 ● | **0.7743±0.0020** | 0.4937±0.0109 ● | 0.4867±0.0120● | 0.5156±0.0000● | **0.7597±0.0023** |
| 5 | 0.6655±0.0022● | 0.6704±0.0000 ● | 0.6707±0.0000 ● | **0.7704±0.0005** | 0.5441±0.0000● | 0.5402±0.0000 ● | 0.5393±0.0000 ● | **0.7587±0.0006** |
| 6 | 0.6625±0.0000● | 0.6621±0.0000● | 0.6606±0.0000 ● | **0.8681±0.0004** | 0.5280±0.0000 ● | 0.5300±0.0000● | 0.5295±0.0000● | **0.8682±0.0004** |
| 7 | 0.2971±0.0005 ● | 0.3038±0.0007● | 0.2848±0.0037 ● | **0.4303±0.0010** | 0.1662±0.0006● | 0.1740±0.0012 ● | 0.1646±0.0031● | **0.3900±0.0011** |
| 8 | 0.1410±0.0030 ● | 0.1586±0.0035 ● | 0.1546±0.0015 ● | **0.7498±0.0056** | 0.0589±0.0027 ● | 0.0679±0.0024● | 0.0718±0.0007 ● | **0.7478±0.0060** |
| 9 | 0.3672±0.0018 ● | 0.3542±0.0021● | 0.3719±0.0015 ● | **0.5736±0.0007** | 0.2213±0.0035 ● | 0.2245±0.0020 ● | 0.2396±0.0030 ● | **0.5302±0.0010** |
| 10 | 0.8489±0.0000● | 0.8489±0.0000 ● | 0.8489±0.0000● | **0.9224±0.0002** | 0.7795±0.0000● | 0.7795±0.0000 ● | 0.7795±0.0000● | **0.9237±0.0001** |
| 11 | **0.9711±0.0000** | **0.9711±0.0000** | **0.9711±0.0000** | 0.9707±0.0000 | 0.9568±0.0000 | 0.9568±0.0000 | 0.9568±0.0000 | 0.9576±0.0000 |
| 12 | 0.9497±0.0000 | 0.9517±0.0000 | 0.9482±0.0000 ● | **0.9534±0.0000** | 0.9496±0.0000 | 0.9517±0.0000 | 0.9481±0.0000 ● | **0.9534±0.0000** |
| 13 | 0.9474±0.0000● | 0.9474±0.0000● | 0.9474±0.0000● | **0.9808±0.0000** | 0.9218±0.0000● | 0.9218±0.0000 ● | 0.9218±0.0000 ● | **0.9808±0.0000** |
| 14 | 0.9484±0.0000● | 0.9484±0.0000● | 0.9484±0.0000 ● | **0.9868±0.0000** | 0.9232±0.0000● | 0.9232±0.0000● | 0.9232±0.0000 ● | **0.9867±0.0000** |
| 15 | 0.5381±0.0006 ● | 0.5527±0.0007● | 0.5409±0.0004 ● | **0.6880±0.0000** | 0.4252±0.0064● | 0.4785±0.0075● | 0.4450±0.0059● | **0.6875±0.0001** |
| 16 | 0.9371±0.0000● | 0.9395±0.0000 ● | 0.9371±0.0000● | **0.9864±0.0000** | 0.9066±0.0000● | 0.9113±0.0000 ● | 0.9066±0.0000● | **0.9864±0.0000** |
| 17 | 0.9470±0.0002 ● | 0.9518±0.0001● | 0.9466±0.0001● | **0.9708±0.0000** | 0.9470±0.0002 ● | 0.9517±0.0001● | 0.9466±0.0001● | **0.9708±0.0000** |
| 18 | 0.9895±0.0000● | 0.9880±0.0001● | 0.9909±0.0000● | **1.0000±0.0000** | 0.9895±0.0000● | 0.9880±0.0001● | 0.9909±0.0000● | **1.0000±0.0000** |
| 19 | 0.8203±0.0039● | 0.8102±0.0080● | 0.8296±0.0070● | **0.9664±0.0003** | 0.8075±0.0042● | 0.7964±0.0092● | 0.8165±0.0081 ● | **0.9661±0.0003** |
| 20 | 0.8442±0.0000● | 0.8421±0.0000● | 0.8432±0.0000● | **0.8485±0.0000** | 0.8389±0.0000● | 0.8368±0.0000● | 0.8379±0.0000 | **0.8421±0.0000** |
| 21 | 0.0319±0.0001● | 0.0271±0.0001 ● | 0.0276±0.0001● | **0.4207±0.0033** | 0.0080±0.0000● | 0.0062±0.0000● | 0.0081±0.0000● | **0.3996±0.0039** |
| 22 | 0.4637±0.0171● | 0.4853±0.0082● | 0.4446±0.0045● | **0.8485±0.0004** | 0.4112±0.0216● | 0.4318±0.0114● | 0.3897±0.0047● | **0.8484±0.0004** |
| 23 | 0.4541±0.0106 ● | 0.5035±0.0072● | 0.4074±0.0160● | **0.9519±0.0005** | 0.4194±0.0144 ● | 0.4706±0.0103● | 0.3646±0.0207● | **0.9518±0.0005** |
| 24 | 0.1803±0.0023● | 0.1522±0.0026● | 0.1514±0.0021● | **0.3499±0.0008** | 0.1030±0.0007 ● | 0.0947±0.0012● | 0.0901±0.0008● | **0.3108±0.0009** |
| 25 | 0.9205±0.0035● | 0.9156±0.0048● | 0.9509±0.0004 ● | **0.9717±0.0000** | 0.9167±0.0043● | 0.9108±0.0063● | 0.9509±0.0004 ● | **0.9717±0.0000** |

into two parts based on different evaluation measures. The lowest value of each part in each row is underlined. Euclidean distances ($d_{ED}$) and information entropies ($d_{IE}$) are defined:

$$d_{ED}(\boldsymbol{A}, \boldsymbol{B}) = \sqrt{\sum_{i=1}^{n}\sum_{i=1}^{n}(a_{ij} - b_{ij})^2}, \tag{87}$$

$$d_{IE}(\boldsymbol{A}) = \sum_{r=1}^{k}(\sum_{i=1}^{m_r}\sum_{j=1}^{m_r} p_{ij}log(p_{ij})), \tag{88}$$

where $a_{ij}$ and $b_{ij}$ are the elements in the i-th row and j-th column of matrices $\boldsymbol{A}$ and $\boldsymbol{B}$, respectively and $p_{ij}$ is the probability value of the $i$-th row and $j$-th column element in matrix $\boldsymbol{A}_{[r][r]}$. Figure 14 to 17 presents the t-SNE of the similarity matrices of the feature representations for each method. As shown in the tables and figures, the CE-PSED method demonstrates superior performance in feature representation and class discrimination.

### 11.1.6. COMPARISON WITH BASELINE METHODS

In this section, we present a comparative analysis of the CE-PSED-based method against several existing baseline approaches across five image datasets, as summarized in Table 11. The results demonstrate that the CE-PSED-based method consistently achieves superior accuracy compared to the baseline methods, highlighting its effectiveness in diverse scenarios.

### 11.1.7. COMPARISON OF DIFFERENT NETWORK STRUCTURES

In addition, we use loss functions to train more complex networks. The experimental results and detailed parameter configuration are as follows. In this study, we conducted experiments using hardware configurations including Intel (R) Core (TM) i7-14700F CPU, 16GB RAM, and NVIDIA GeForce RTX 4060 GPU. The experiment was conducted on the Windows operating system, with Python 3.10 as the programming language and PyTorch 2.4 library for model development and training.

For training the Visual Transformer (ViT) (Dosovitskiy et al., 2020) model, we use a stochastic gradient descent (SGD)

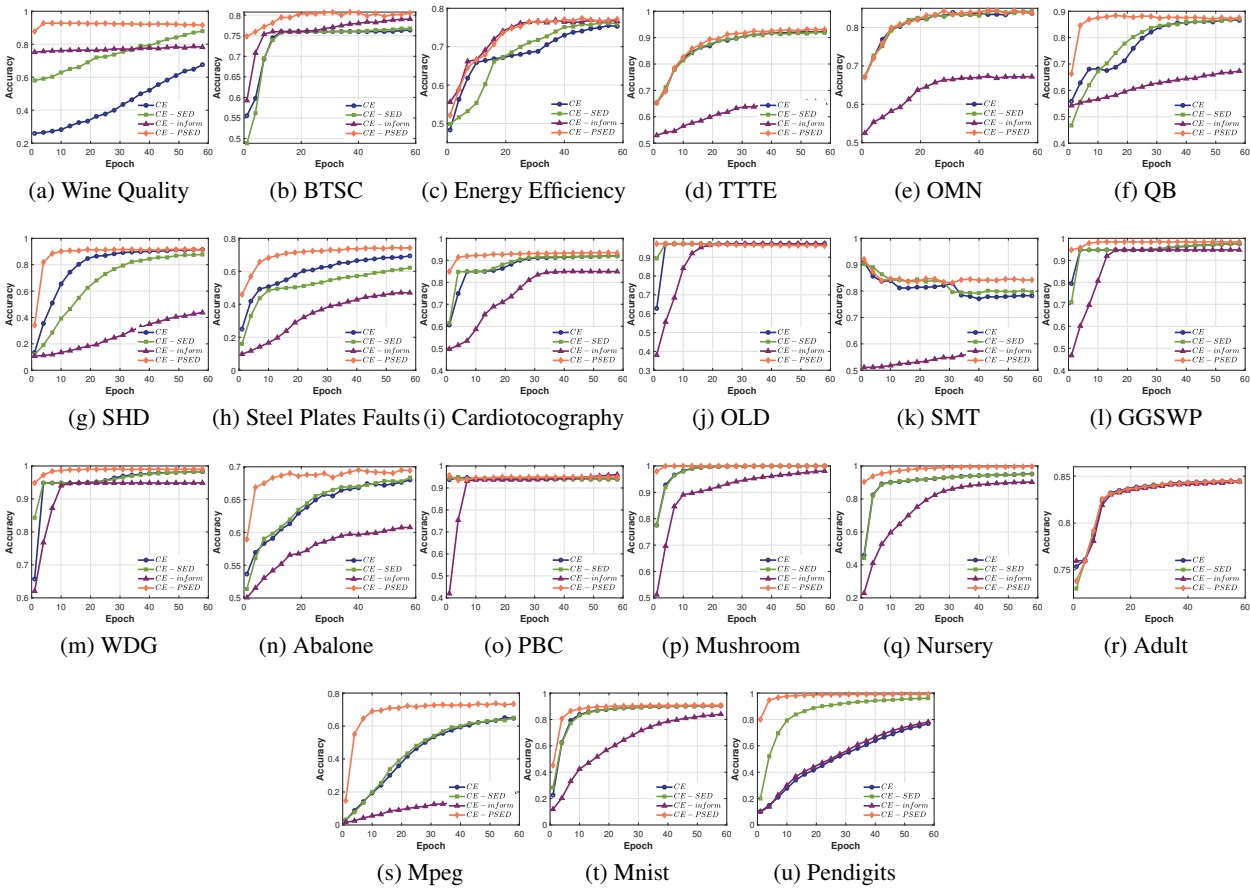

*Figure 11.* Accuracy curves based on different loss functions when model layers is 3.

optimizer with a learning rate set to 0.001. The batch size was set to 64, and the model was trained for a total of 100 iteration cycles. The dataset is divided into training and testing subsets in a ratio of 7:3. Similarly, the ConvNeXt (Liu et al., 2022) model was trained using the AdamW optimizer with a learning rate set to 0.004. The model also uses a batch size of 64 and is trained over 100 iteration cycles. Consistent with the ViT model, the dataset is divided into training and testing sets, maintaining the same 7:3 ratio.

The performance evaluation of both models is conducted using accuracy and F-measure as the main measures. As shown in the table 12, the analysis of the results indicates that our proposed method has made significant improvements compared to existing methods. This demonstrates the effectiveness of our method.

### 11.1.8. COMPARISON OF RUNTIME FOR DIFFERENT LOSS FUNCTIONS

We conduct a runtime comparison, as shown in the table 13. For the first 20 benchmark datasets, the table presents the total time consumption (in seconds) for both training and testing. For image datasets, the recorded values represent the training and prediction time (in seconds) on the fully connected network shown in Figure 3, after feature extraction using either MoCo v3 or VGG. From the table 13, it can be observed that our method does not significantly increase computational time.

*Table 8.* Euclidean distance and information entropy between similarity matrix and $\mathbf{YY}^T$ based on different loss functions when model layers is 3.

| Data | $d_{ED}$ | | | | $d_{IE}$ | | | |
|---|---|---|---|---|---|---|---|---|
| | **CE** | **CE-SED** | **CE-inform** | **CE-PSED** | **CE** | **CE-SED** | **CE-inform** | **CE-PSED** |
| 1 | 76.1735 | 76.1735 | 77.5304 | 76.5485 | 37.0368 | 37.0368 | 37.0385 | 37.0500 |
| 2 | 118.0135 | 118.0135 | 116.3866 | 106.0565 | 35.9276 | 35.9276 | 35.9308 | 35.8186 |
| 3 | 123.2038 | 123.2038 | 129.5480 | 110.1027 | 35.4940 | 35.4940 | 35.5365 | 35.2933 |
| 4 | 162.3454 | 162.3454 | 159.9709 | 149.7737 | 37.2954 | 37.2954 | 37.3005 | 37.2043 |
| 5 | 173.1156 | 173.1156 | 175.2450 | 154.3241 | 37.9086 | 37.9086 | 37.8876 | 37.7090 |
| 6 | 166.4008 | 166.4008 | 169.8654 | 135.0187 | 38.0959 | 38.0959 | 38.1075 | 37.9415 |
| 7 | 329.6840 | 329.6840 | 332.4945 | 260.9751 | 608.8749 | 608.8749 | 608.8462 | 607.8717 |
| 8 | 257.0381 | 257.0381 | 248.1448 | 243.9900 | 541.0715 | 541.0715 | 539.5068 | 542.3536 |
| 9 | 399.3402 | 399.3402 | 390.1124 | 313.9013 | 359.1309 | 359.1309 | 359.0130 | 358.5208 |
| 10 | 254.0633 | 254.0633 | 247.5812 | 209.8767 | 44.2149 | 44.2149 | 44.2136 | 44.0434 |
| 11 | 189.9269 | 189.9269 | 187.7281 | 180.3647 | 47.2859 | 47.2859 | 47.2887 | 47.2785 |
| 12 | 279.1663 | 279.1663 | 286.4833 | 276.5521 | 46.8595 | 46.8595 | 46.8014 | 46.8092 |
| 13 | 311.5175 | 311.5175 | 309.4672 | 205.7426 | 49.3400 | 49.3400 | 49.3379 | 49.2880 |
| 14 | 336.9726 | 336.9726 | 335.6694 | 219.6362 | 50.0835 | 50.0835 | 50.0865 | 50.0487 |
| 15 | 466.5055 | 466.5055 | 465.5321 | 444.3993 | 45.1351 | 45.1351 | 45.0850 | 45.2805 |
| 16 | 338.2421 | 338.2421 | 381.2186 | 295.4883 | 52.8144 | 52.8144 | 52.8225 | 52.8146 |
| 17 | 895.9040 | 895.9040 | 845.0391 | 442.9792 | 53.2286 | 53.2286 | 53.2484 | 53.2705 |
| 18 | 517.6101 | 517.6101 | 511.4658 | 223.7286 | 54.0381 | 54.0381 | 54.0401 | 54.0677 |
| 19 | 1734.1428 | 1734.1428 | 1674.9242 | 1809.1698 | 201.6555 | 201.6555 | 201.6630 | 201.5141 |
| 20 | 4301.6353 | 4301.6353 | 4233.9531 | 4129.7095 | 66.0202 | 66.0202 | 65.9704 | 65.9258 |
| 21 | 233.7108 | 233.7108 | 239.6036 | 222.7377 | 820.5886 | 820.5886 | 823.8793 | 825.4946 |
| 22 | 1055.0093 | 1055.0093 | 1042.0280 | 919.8159 | 837.9047 | 837.9047 | 837.8022 | 837.8456 |
| 23 | 1187.5291 | 1187.5291 | 1180.5438 | 1043.7703 | 851.2291 | 851.2291 | 851.2600 | 850.7767 |
| 24 | 1381.6508 | 1381.6508 | 1406.4387 | 1391.6364 | 29363.1641 | 29363.1641 | 29390.8047 | 29438.4980 |
| 25 | 1755.2109 | 1755.2109 | 1602.9089 | 1572.4526 | 962.3068 | 962.3068 | 962.1885 | 962.1676 |

*Table 9.* Euclidean distance and information entropy between similarity matrix and $\mathbf{YY}^T$ based on different loss functions when model layers is 5.

| Data | $d_{ED}$ | | | | $d_{IE}$ | | | |
|---|---|---|---|---|---|---|---|---|
| | **CE** | **CE-SED** | **CE-inform** | **CE-PSED** | **CE** | **CE-SED** | **CE-inform** | **CE-PSED** |
| 1 | 79.6850 | 78.5153 | 78.1867 | 78.6162 | 37.0328 | 37.0405 | 37.0459 | 37.0448 |
| 2 | 121.9238 | 124.8685 | 125.6604 | 113.0838 | 35.9199 | 35.9311 | 35.9453 | 35.9171 |
| 3 | 139.6086 | 139.7598 | 139.5939 | 112.8470 | 35.5393 | 35.5488 | 35.5446 | 35.4087 |
| 4 | 176.6676 | 177.8770 | 173.6018 | 142.6622 | 37.3264 | 37.2951 | 37.3265 | 37.2480 |
| 5 | 184.9006 | 180.9771 | 184.6760 | 152.6356 | 37.9203 | 37.9225 | 37.9254 | 37.7662 |
| 6 | 181.3291 | 186.9172 | 183.1280 | 141.9453 | 38.1332 | 38.1379 | 38.1374 | 37.9251 |
| 7 | 351.0292 | 350.8264 | 353.8266 | 270.3388 | 608.8852 | 609.0336 | 608.8508 | 607.8669 |
| 8 | 287.1577 | 311.6756 | 321.5478 | 264.3753 | 541.0257 | 541.8483 | 542.0345 | 542.2803 |
| 9 | 451.0792 | 447.5596 | 432.1057 | 332.7613 | 359.3900 | 359.3643 | 359.3124 | 358.7696 |
| 10 | 272.9623 | 273.0852 | 270.0412 | 203.4734 | 44.2175 | 44.2134 | 44.2204 | 44.1212 |
| 11 | 184.7525 | 186.9483 | 185.5223 | 201.9718 | 47.2881 | 47.2837 | 47.2859 | 47.2416 |
| 12 | 269.4701 | 295.2358 | 286.0780 | 272.2967 | 46.8527 | 46.8941 | 46.8962 | 46.8753 |
| 13 | 314.9090 | 308.2535 | 310.5731 | 216.3729 | 49.3310 | 49.3399 | 49.3396 | 49.2533 |
| 14 | 338.2912 | 337.2207 | 335.4204 | 214.1152 | 50.0805 | 50.0874 | 50.0830 | 50.0418 |
| 15 | 484.8264 | 475.2573 | 476.4166 | 452.1942 | 45.3697 | 45.3445 | 45.2689 | 45.1519 |
| 16 | 370.6945 | 354.5944 | 505.9410 | 297.0087 | 52.8071 | 52.8160 | 52.8222 | 52.8105 |
| 17 | 829.2169 | 910.4088 | 802.9050 | 461.3801 | 53.2784 | 53.2987 | 53.2917 | 53.2539 |
| 18 | 486.7239 | 527.8088 | 569.7032 | 245.3424 | 54.0134 | 54.0535 | 53.9989 | 54.0640 |
| 19 | 1907.5442 | 1875.5889 | 1901.7069 | 2104.1467 | 201.6993 | 201.6781 | 201.6916 | 201.6814 |
| 20 | 4519.4912 | 4370.4189 | 4409.1572 | 4183.5938 | 66.0612 | 66.0230 | 66.0429 | 65.9723 |
| 21 | 284.0164 | 279.8763 | 263.6331 | 238.7444 | 826.9400 | 827.9311 | 825.7845 | 825.6633 |
| 22 | 1183.8347 | 1212.5712 | 1279.2507 | 1003.7283 | 838.1042 | 838.2512 | 837.7303 | 837.8041 |
| 23 | 1321.9459 | 1382.4373 | 1364.1901 | 1143.9182 | 851.9315 | 852.3873 | 852.1819 | 852.2212 |
| 24 | 1596.7644 | 1618.1927 | 1563.8016 | 1454.6886 | 29456.8047 | 29419.6973 | 29415.5566 | 29453.1133 |
| 25 | 2040.1495 | 2058.1721 | 2063.2100 | 1792.5488 | 962.3513 | 962.1644 | 962.4064 | 962.1599 |

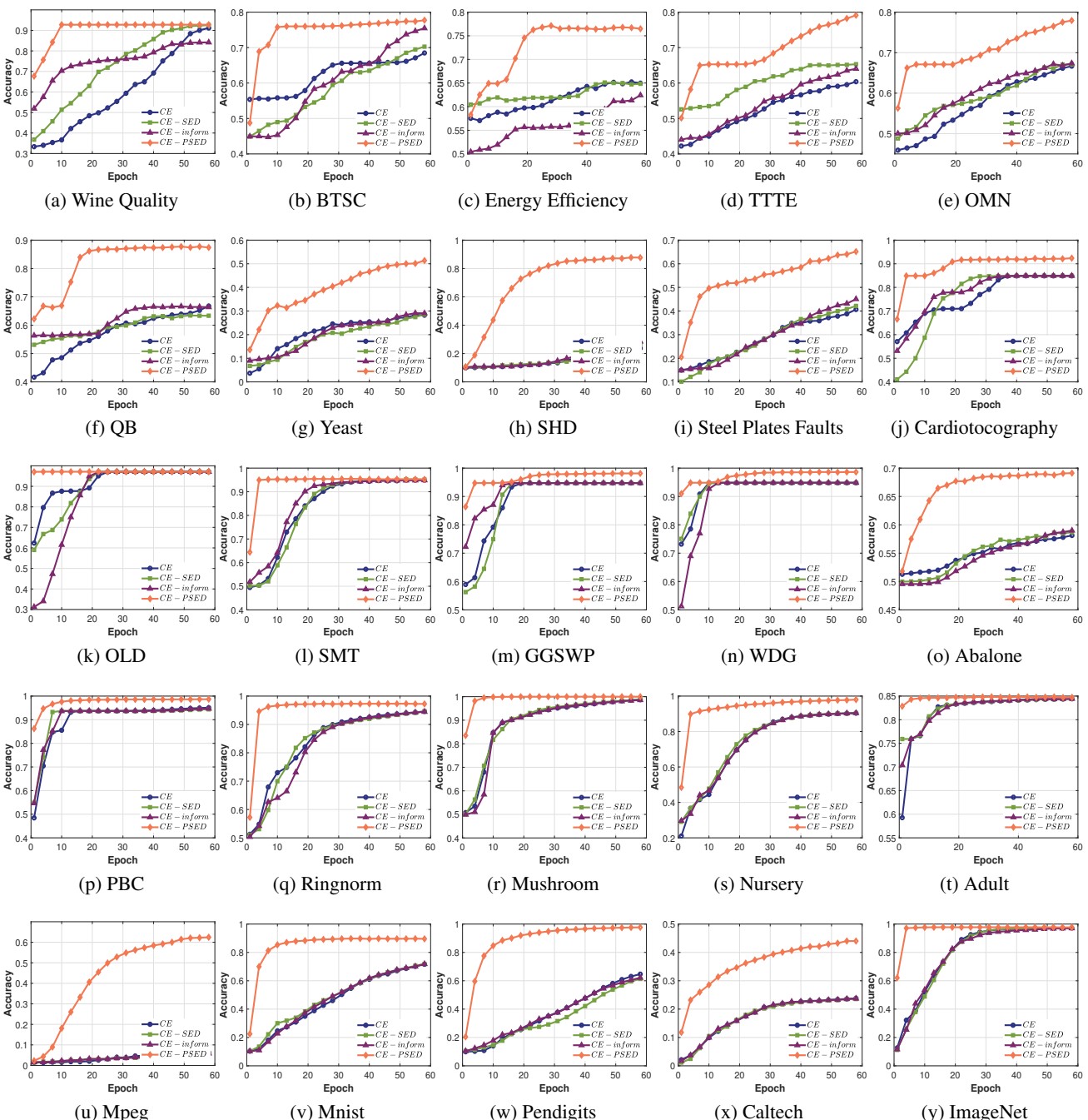

*Figure 12.* Accuracy curves based on different loss functions when model layers is 5.

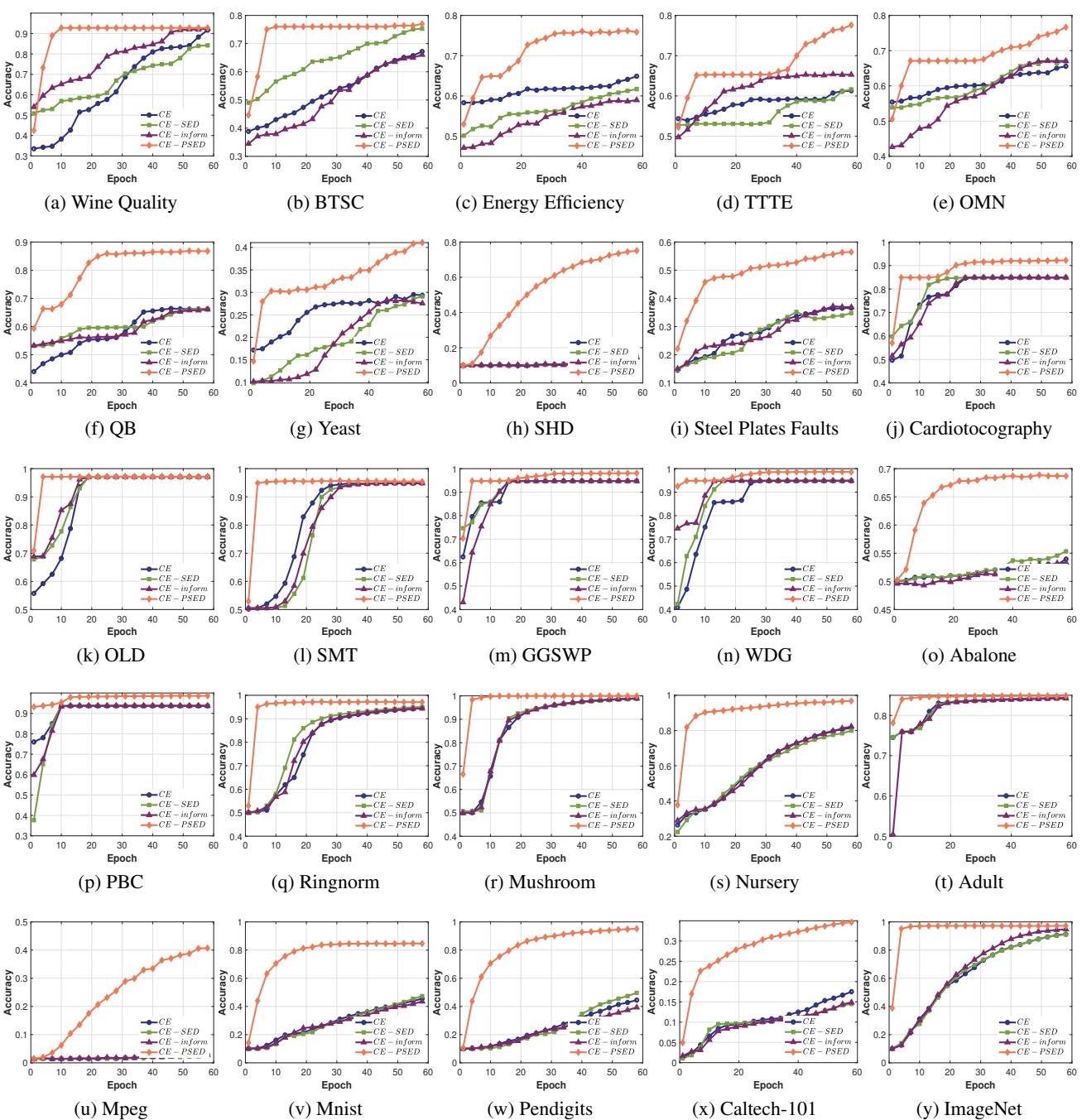

*Figure 13.* Accuracy curves based on different loss functions when model layers is 8.

*Table 10.* Euclidean distance and information entropy between similarity matrix and $\mathbf{Y}\mathbf{Y}^T$ based on different loss functions when model layers is 8.

| Data | $d_{ED}$ | | | | $d_{IE}$ | | | |
|---|---|---|---|---|---|---|---|---|
| | **CE** | **CE-SED** | **CE-inform** | **CE-PSED** | **CE** | **CE-SED** | **CE-inform** | **CE-PSED** |
| 1 | 78.5572 | 77.6711 | 77.5050 | 76.6356 | 37.0452 | 37.0489 | 37.0449 | 37.0509 |
| 2 | 123.3103 | 124.2150 | 124.3540 | 119.5258 | 35.9527 | 35.9524 | 35.9383 | 35.9547 |
| 3 | 143.4134 | 142.9109 | 142.7595 | 125.0313 | 35.5476 | 35.5408 | 35.5183 | 35.5139 |
| 4 | 177.0358 | 175.9595 | 177.6065 | 144.1926 | 37.3276 | 37.3231 | 37.2873 | 37.1813 |
| 5 | 184.6225 | 186.0866 | 187.0768 | 158.6200 | 37.9243 | 37.9152 | 37.9159 | 37.8397 |
| 6 | 192.7056 | 191.0630 | 194.8601 | 147.8844 | 38.1393 | 38.1397 | 38.1393 | 37.6394 |
| 7 | 351.8983 | 354.9803 | 352.5111 | 307.3115 | 609.0768 | 609.0687 | 609.1129 | 608.3528 |
| 8 | 397.7230 | 347.8330 | 399.9164 | 256.9694 | 542.7438 | 542.2402 | 542.5559 | 541.8681 |
| 9 | 460.3625 | 464.0630 | 466.8538 | 362.0203 | 359.4289 | 359.3731 | 359.4217 | 358.4846 |
| 10 | 273.0928 | 272.3560 | 276.0269 | 206.4414 | 44.2209 | 44.2218 | 44.2104 | 44.0273 |
| 11 | 184.6531 | 188.8407 | 187.7007 | 182.5056 | 47.2881 | 47.2871 | 47.2887 | 47.2832 |
| 12 | 368.6120 | 332.2215 | 391.8640 | 298.4601 | 46.9220 | 46.9126 | 46.9306 | 46.8262 |
| 13 | 312.4539 | 312.4441 | 311.5305 | 185.7752 | 49.3347 | 49.3400 | 49.3412 | 49.3174 |
| 14 | 335.2605 | 333.6607 | 335.5419 | 203.2962 | 50.0845 | 50.0865 | 50.0830 | 50.0606 |
| 15 | 510.7383 | 517.9683 | 500.6602 | 469.6039 | 45.4402 | 45.4203 | 45.4393 | 45.3898 |
| 16 | 516.2045 | 515.2233 | 500.0887 | 297.2825 | 52.8287 | 52.8268 | 52.8293 | 52.7993 |
| 17 | 941.3668 | 688.6042 | 876.5587 | 484.4434 | 53.2947 | 53.2633 | 53.2934 | 53.2362 |
| 18 | 655.9843 | 511.2377 | 670.1942 | 416.2432 | 54.0622 | 54.0526 | 54.0665 | 54.0659 |
| 19 | 2047.2732 | 2357.8250 | 1974.6055 | 2180.1816 | 201.6154 | 201.7070 | 201.6311 | 201.6664 |
| 20 | 4545.1484 | 4479.0952 | 4472.4590 | 4337.5742 | 66.0634 | 66.0526 | 66.0532 | 65.9778 |
| 21 | 313.1884 | 294.9319 | 302.1780 | 243.5456 | 829.1549 | 820.5233 | 825.6597 | 815.4856 |
| 22 | 1374.4672 | 1269.7419 | 1305.0961 | 1149.5310 | 836.5143 | 836.2145 | 838.1793 | 837.6033 |
| 23 | 1461.3339 | 1477.7767 | 1404.8453 | 1165.8860 | 852.1779 | 851.9873 | 852.3888 | 852.2087 |
| 24 | 2053.9851 | 1847.4176 | 1824.1750 | 1530.5894 | 29493.0820 | 29490.4590 | 29429.0820 | 29470.7949 |
| 25 | 2384.8987 | 2204.3345 | 2293.9170 | 1886.0833 | 961.9738 | 962.0734 | 962.1173 | 961.9856 |

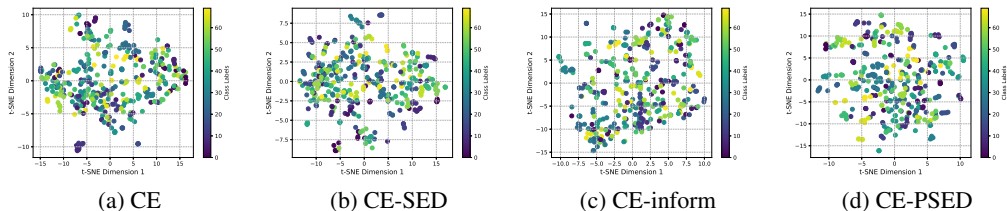

| (a) CE | (b) CE-SED | (c) CE-inform | (d) CE-PSED |

*Figure 14.* The t-SNE of Mpeg.

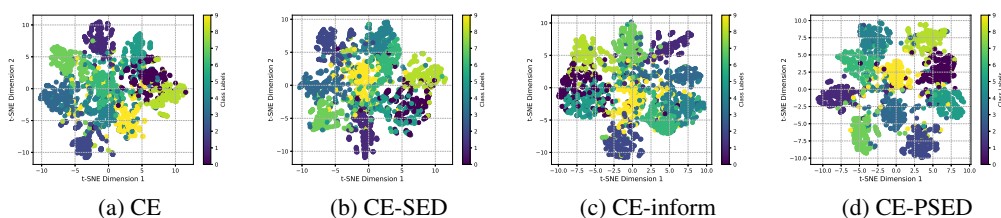

| (a) CE | (b) CE-SED | (c) CE-inform | (d) CE-PSED |

*Figure 15.* The t-SNE of Mnist.

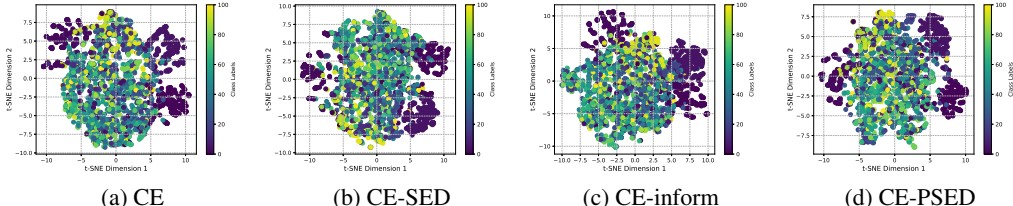

(a) CE       (b) CE-SED       (c) CE-inform       (d) CE-PSED

*Figure 16.* The t-SNE of Caltech-101.

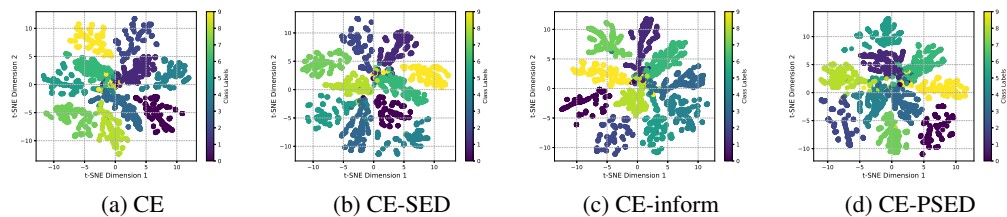

(a) CE       (b) CE-SED       (c) CE-inform       (d) CE-PSED

*Figure 17.* The t-SNE of ImageNet.

*Table 11.* Comparison with baseline methods in Accuracy

| Data | Baseline Citation | Accuracy | CE-PSED |
|---|---|---|---|
| Mpeg | (Bai et al., 2009) | 0.5200 | 0.7338 |
| | (Grigorescu & Petkov, 2003) | 0.5000 | |
| Mnist(6996 pcs) | (Ren et al., 2016) | 0.7946 | |
| | (He & Sun, 2015) | 0.8073 | |
| | (Goodfellow et al., 2013) | 0.8257 | 0.9062 |
| | (Hinton et al., 1999) | 0.8768 | |
| Mnist(70000 pcs) | (Byerly et al., 2021) | 0.9987 | |
| Pendigits | (McConville et al., 2021) | 0.8850 | |
| | (Li et al., 2021) | 0.8227 | |
| | (Toth & Oberhauser, 2020) | 0.9550 | 0.9908 |
| | (Cai & Chen, 2015) | 0.8155 | |
| | (van der Maaten & Hinton, 2008) | 0.8930 | |
| Caltech-101 | (Bansal et al., 2021) | 0.4500 | |
| | (Bansal et al., 2021) | 0.4400 | 0.5005 |
| | (Chen & Guestrin, 2016) | 0.5000 | |
| | (Irle & Kauschke, 2011) | 0.5000 | |
| ImageNet | (Chen et al., 2023) | 0.9094 | |
| | (Yu et al., 2022) | 0.9100 | 0.9762 |
| | (Wortsman et al., 2022) | 0.9098 | |
| | (Pham et al., 2021) | 0.9020 | |

Table 12. Performance Comparison between Network Structures

| Model | Dataset | Accuracy | | F-measure | |
|---|---|---|---|---|---|
| | | Original | PSED | Original | PSED |
| **ConvNeXt** | Mpeg | 0.8238 | **0.8310** | 0.8170 | **0.8280** |
| | Mnist | 0.9904 | **0.9925** | 0.9900 | **0.9930** |
| | Pendigits | 0.9708 | **0.9832** | 0.9699 | **0.9826** |
| | Caltech-101 | 0.6429 | **0.6536** | 0.6100 | **0.6230** |
| | ImageNet | 0.9651 | **0.9687** | 0.9650 | **0.9695** |
| **ViT** | Mpeg | 0.6820 | **0.7095** | 0.5515 | **0.6811** |
| | Mnist | 0.8590 | **0.9290** | 0.8550 | **0.9277** |
| | Pendigits | 0.9568 | **0.9711** | 0.9512 | **0.9700** |
| | Caltech-101 | 0.7393 | **0.8500** | 0.6034 | **0.7866** |
| | ImageNet | 0.9964 | **0.9968** | 0.9952 | **0.9973** |

Table 13. Comparison of runtime (seconds) for different loss functions

| Dataset | CE | CE-SED | CE-inform | CE-PSED |
|---|---|---|---|---|
| 1 | 2.34 | 3.07 | 3.49 | 6.03 |
| 2 | 2.35 | 3.07 | 3.44 | 8.94 |
| 3 | 2.44 | 3.93 | 3.57 | 9.14 |
| 4 | 2.88 | 8.70 | 4.47 | 11.13 |
| 5 | 3.23 | 9.41 | 4.83 | 12.02 |
| 6 | 3.33 | 9.65 | 5.03 | 6.46 |
| 7 | 6.44 | 13.33 | 12.58 | 8.96 |
| 8 | 12.81 | 10.06 | 27.73 | 12.09 |
| 9 | 13.75 | 7.62 | 18.54 | 11.58 |
| 10 | 13.72 | 7.62 | 8.96 | 9.31 |
| 11 | 10.30 | 10.25 | 11.59 | 11.81 |
| 12 | 9.99 | 12.97 | 15.32 | 15.93 |
| 13 | 9.98 | 12.94 | 14.90 | 15.39 |
| 14 | 10.93 | 14.20 | 16.35 | 27.15 |
| 15 | 8.29 | 10.76 | 12.74 | 13.26 |
| 16 | 15.52 | 30.26 | 33.69 | 24.15 |
| 17 | 22.06 | 28.64 | 33.89 | 35.15 |
| 18 | 34.30 | 31.46 | 37.09 | 38.56 |
| 19 | 38.57 | 49.95 | 63.52 | 68.38 |
| 20 | 96.47 | 126.55 | 149.74 | 155.72 |
| Mpeg | 8.18 | 9.94 | 16.36 | 18.43 |
| Mnist | 27.39 | 35.20 | 51.52 | 55.51 |
| Pendigits | 22.40 | 29.61 | 45.99 | 50.55 |
| Caltech-101 | 32.00 | 42.59 | 70.91 | 80.32 |
| ImageNet | 46.86 | 58.76 | 86.41 | 96.38 |

