# OpenReview forum: "Stabilizing Sample Similarity in Representation via Mitigating Random Consistency"
_ICML.cc/2025/Conference — ICML 2025 poster_

### Official Review · Reviewer_WSCE · 2025-03-01

**Overall Recommendation:** 3

**Summary:**

This paper addresses the challenge of measuring and improving representation quality in deep learning models. The authors propose a novel loss function called Pure Square Euclidean Distance (PSED) that measures the discriminative ability of representations by computing the Euclidean distance between a similarity matrix and the true adjacency matrix.

**Claims And Evidence:**

Yes

**Essential References Not Discussed:**

No

**Experimental Designs Or Analyses:**

Yes

**Methods And Evaluation Criteria:**

Yes, the experimental evaluation is extensive, covering both benchmark and image datasets, and it demonstrates the method's effectiveness.

**Other Comments Or Suggestions:**

N/A

**Other Strengths And Weaknesses:**

Weakness:

1. The experiments focus primarily on fully connected networks instead of exploring more complex architectures, such as CNNs or Transformers, which are common in modern deep learning.
2. PSED still introduces additional computational complexity compared to the standard CE loss, especially for large batch size training. Additional computation cost analysis should be conducted.

**Questions For Authors:**

N/A

**Relation To Broader Scientific Literature:**

N/A

**Theoretical Claims:**

Yes, The paper provides solid mathematical analysis of PSED's properties, including formal proofs of heterogeneity and unbiasedness, and establishes generalization performance bounds using exponential Orlicz norm-based concentration inequalities.

---

> ### Author Rebuttal · Authors · 2025-03-31
>
> Dear Reviewer WSCE,
>
> &nbsp;&nbsp;&nbsp;&nbsp; We are very grateful for your valuable comments and questions. The responses are as follows:
>
> **Response to Weaknesses 1:**
>
> &nbsp;&nbsp;&nbsp;&nbsp; For the image dataset, we solely employ VGG and MoCo v3 for feature extraction. The specific methodology is detailed in Section 7.2 of the main text and Appendix 3.0.1. In this modification, we trained more sophisticated networks (ViT and CNN) using our loss function. The experimental results and detailed parameter configurations are as follows.
>
> &nbsp;&nbsp;&nbsp;&nbsp; In this study, we conduct experiments utilizing a hardware configuration that includes an Intel (R) Core (TM) i7-14700F CPU, 16GB of RAM, and an NVIDIA GeForce RTX 4060 GPU. The experiments are carried out on a Windows operating system, with Python 3.10 serving as the programming language and the PyTorch 2.4 library employed for model development and training.
>
> &nbsp;&nbsp;&nbsp;&nbsp; For the Vision Transformer (ViT) [1] model training, we use a stochastic gradient descent (SGD) optimizer with a learning rate set to 0.001. The batch size is set to 64, and the model is trained for a total of 100 epochs. The dataset is partitioned into training and testing subsets with a ratio of 7:3.
>
> &nbsp;&nbsp;&nbsp;&nbsp; Similarly, the ConvNet [2] model employs the AdamW optimizer, with the learning rate set to 0.004. This model also uses a batch size of 64 and undergoes training for 100 epochs. In alignment with the ViT model, the dataset is divided into training and testing sets, preserving the same 7:3 ratio.
>
> &nbsp;&nbsp;&nbsp;&nbsp; Performance evaluation of both models is conducted using accuracy and F-measure as the primary metrics. Analysis of the results, as presented in the table, our method shows better performance over existing approaches. This indicates the effectiveness of our method.
>
> | Method| ConvNet| ConvNet-PSED | ViT| ViT-PSED |
> |- |- |- |- |- |
> |**Accuracy**	|	 	|	 	|	 	|	 	|
> |Mpeg 	|	0.8238	|	**0.8310**	|	0.6820	|	**0.7095**	|
> |Mnist |	0.9904	|	**0.9925**	|	0.8590	|	**0.9290**	|
> |Pendigits |	0.9708	|	**0.9832**	|	0.9568	|	**0.9711**	|
> |Caltech-101   	|	0.6429	|	**0.6536**	|	0.7393	|	**0.8500**	|
> |ImageNet       	|	0.9651	|	**0.9687**	|	0.9964	|	**0.9968**	|
> |**F-measure**	|	 	|	 	|	 	|	 	|
> |Mpeg	|	0.8170	|	**0.8280**	|	0.5515	|	**0.6811**	|
> |Mnist |	0.9900	|	**0.9930**	|	0.8550	|	**0.9277**	|
> |Pendigits  |	0.9699	|	**0.9826**	|	0.9512	|	**0.9700**	|
> |Caltech-101   	|	0.6100	|	**0.6230**	|	0.6034	|	**0.7866**	|
> |ImageNet       	|	0.9650	|	**0.9695**	|	0.9952	|	**0.9973**	|
>
> [1] Alexey Dosovitskiy, Lucas Beyer, Alexander Kolesnikov, et al. An Image is Worth 16x16 Words: Transformers for Image Recognition at Scale. In International Conference on Learning Representations, 2021.
>
> [2] Zhuang Liu, Hanzi Mao, Chao-Yuan Wu, et al. A ConvNet for the 2020s. In 2022 IEEE/CVF Conference on Computer Vision and Pattern Recognition, pages 11966–11976, 2022.
>
> **Response to Weaknesses 2:**
>
> &nbsp;&nbsp;&nbsp;&nbsp; Thank you for your suggestions. We have added a runtime comparison, as shown in the table below. In this study, we conduct experiments utilizing a hardware configuration that includes an Intel (R) Core (TM) i7-12700F CPU, 32GB of RAM, and an NVIDIA GeForce RTX 3060 Ti GPU. For the first 20 benchmark datasets, the table below presents the total time consumption (in seconds) for both training and testing. For image datasets, the recorded values represent the training and prediction time (in seconds) on the fully connected network shown in Figure 3, after feature extraction using either MoCo v3 or VGG. From the table, it can be observed that our method does not significantly increase computational time.
>
> |	Dataset	|	CE	|	CE-SED	|	CE-inform	|	CE-PSED	|
> |--|--|--|--|--|
> |	1	|	2.34	|	3.07	|	3.49	|	6.03	|
> |	2	|	2.35	|	3.07	|	3.44	|	8.94	|
> |	3	|	2.44	|	3.93	|	3.57	|	9.14	|
> |	4	|	2.88	|	8.7	|	4.47	|	11.13	|
> |	5	|	3.23	|	9.41	|	4.83	|	12.02	|
> |	6	|	3.33	|	9.65	|	5.03	|	6.46	|
> |	7	|	6.44	|	13.33	|	12.58	|	8.96	|
> |	8	|	12.81	|	10.06	|	27.73	|	12.09	|
> |	9	|	13.75	|	7.62	|	18.54	|	11.58	|
> |	10	|	13.72	|	7.62	|	8.96	|	9.31	|
> |	11	|	10.3	|	10.25	|	11.59	|	11.81	|
> |	12	|	9.99	|	12.97	|	15.32	|	15.93	|
> |	13	|	9.98	|	12.94	|	14.9	|	15.39	|
> |	14	|	10.93	|	14.2	|	16.35	|	27.15	|
> |	15	|	8.29	|	10.76	|	12.74	|	13.26	|
> |	16	|	15.52	|	30.26	|	33.69	|	24.15	|
> |	17	|	22.06	|	28.64	|	33.89	|	35.15	|
> |	18	|	34.3	|	31.46	|	37.09	|	38.56	|
> |	19	|	38.57	|	49.95	|	63.52	|	68.38	|
> |	20	|	96.47	|	126.55	|	149.74	|	155.72	|
> |	Mpeg	|	8.18	|	9.94	|	16.36	|	18.43	|
> |	Mnist	|	27.39	|	35.2	|	51.52	|	55.51	|
> |	Pendigits	|	22.4	|	29.61	|	45.99	|	50.55	|
> |	Caltech-101	|	32	|	42.59	|	70.91	|	80.32	|
> |	ImageNet	|	46.86	|	58.76	|	86.41	|	96.38	|

---

> > ### Comment · Reviewer_WSCE · 2025-04-06
> >
> > Thanks for the clarification; my question is solved, and I will keep my positive score. However, I also encourage the author to explore methods to improve the efficiency of the loss computation process, as the results show that the additional cost is not minimal.

---

### Official Review · Reviewer_GtCV · 2025-03-11

**Overall Recommendation:** 3

**Summary:**

The manuscript introduced a new sample similarity measure. The main difference with the existing sample similarity measure is that the new measure mitigates random consistency. The measure forces class-level discrimination. Several theoretical results regarding the measure have been introduced (quality of stohastic estimation, heterogeneity, unbiasedness). Experiments with real-world datasets show that the measure improves classification quality.

**Claims And Evidence:**

Main claims of the paper are valid (see below).

```
• A loss function for measuring the ability of the representation layer is proposed, and an explicit solution for the loss function in the version of eliminating random consistency is given.
• Through theoretical analysis, the advantages of this metric in heterogeneity and unbiasedness have been demonstrated, and a generalization bound has been provided for the generalization performance of the loss function in fully connected layer network structures.
• A fully connected network classification model based on this loss function was proposed, and the effectiveness of the algorithm was verified through extensive experiments.
```

**Essential References Not Discussed:**

--

**Ethical Review Concerns:**

--

**Experimental Designs Or Analyses:**

Experiment design is mostly correct. Which train/test splits you have used?

**Methods And Evaluation Criteria:**

If I understood correctly, for image datasets like ImageNet, Caltech-101 you used fully-connected network.
This is problematic because convolutional architectures are typically applied for images.

**Other Comments Or Suggestions:**

1. > "Random consistency (RC), resulting from randomness rather than true similarity, often arises in consistency evaluation, particularly with limited samples or noise"

this sentence is not clear. Can you please clarify it?

2. K in eq.3 is not defined
3. line 183, Property 3. Matrices I, J are not defined.
4. lin 276, typo: "symmilaritym atrix"

**Other Strengths And Weaknesses:**

It is interesting that you have improved standard cross-entropy loss.

**Questions For Authors:**

1. CE-PSED metric is never defined explicitly, only CE and PSED losses separately.
2. Why the performance gap between CE and CE-PSED is large for some datasets like Mpeg of Caltech-101 ?
3. Which neural architecture you have used for the ImageNet dataset?

**Relation To Broader Scientific Literature:**

The manuscript is related to a broader literature of representational similarity measures.

**Theoretical Claims:**

No

---

> ### Author Rebuttal · Authors · 2025-03-31
>
> Dear Reviewer GtCV,
>
> &nbsp;&nbsp;&nbsp;&nbsp; We are very grateful for your valuable comments and questions. The responses are as follows:
>
> **Response to Methods And Evaluation Criteria:**
>
> &nbsp;&nbsp;&nbsp;&nbsp; For the image dataset, we firstly employed feature extractors such as MoCo v3 or VGG for processing, with downstream classification processing handled by fully connected layers. The specific methodology is detailed in Section 7.2 of the main text and Appendix 3.0.1. We note that there may be a typesetting oversight in Figure 3, where images were mistakenly labeled as direct inputs. This issue will be corrected in the revised version. Besides, in this modification, we implement our loss function on a more sophisticated network architecture(ViT and CNN). The experimental results and detailed settings are listed in our response to Reviewer WSCE Weakness 1.
>
> **Response to Experimental Designs Or Analyses:**
>
> &nbsp;&nbsp;&nbsp;&nbsp; We adopted random splitting to generate the training and test sets, as described in Appendix 3.0.1.
>
> **Response to Other Comments Or Suggestion 1:**
>
> &nbsp;&nbsp;&nbsp;&nbsp; The consistency metric quantifies the degree of agreement or alignment between different variables. Random Consistency (RC) refers to agreement or similarity that occurs purely by chance, not because of any true underlying pattern or relationship. When the sample size is limited or the noise level is high, the random consistency (RC) between the classifier's predicted labels and the true labels increases [1].  For more details on RC, please refer to Question 3 of Reviewer GtCV.
>
> [1] Jieting Wang, Yuhua Qian, Feijiang Li. Learning with Mitigating Random Consistency from the Accuracy Measure, Machine Learning, 2020, 109: 2247-2281
>
> **Response to Other Comments Or Suggestion 2-4:**
>
> &nbsp;&nbsp;&nbsp;&nbsp; $K$, $I$, and $J$ are defined in Equation (1), though we will include their definitions after each subsequent formula. Thank you for your suggestion. Thank you for pointing these out and we will modify them accordingly.
>
> **Response to Question 1:**
>
> &nbsp;&nbsp;&nbsp;&nbsp; CE-PSED refers to  the summation of CE and PSED. Thank you for pointing this and we will add this in the final version.
>
> **Response to Question 2:**
>
> &nbsp;&nbsp;&nbsp;&nbsp;  This is indeed a profound question. As illustrated in Figures 7 and 9 in the appendix, the t-SNE visualization of cross-entropy (CE) for the MPEG and Caltech-101 datasets fails to reveal clear inter-class structures. By introducing the block-diagonal constraint on the similarity matrix—that is, incorporating PSED as part of the loss function—we can enhance inter-class separation in data representation, thereby improving model performance, shown as Table 7 in Appendix. We will add this explanation in the final version.
>
> **Response to Question 3:**
>
> &nbsp;&nbsp;&nbsp;&nbsp; For the image datasets, we firstly employed MoCo v3 or VGG for feature extracting, with downstream classification processing handled by fully connected layers. The specific methodology is detailed in Section 7.2 of the main text and Appendix 3.0.1.

---

> > ### Comment · Reviewer_GtCV · 2025-04-07
> >
> > Thank you for the clarifications. The clarifications should be included into the main part of the paper.
> > I prefer to remain my evaluation unchanged (weak accept).

---

### Official Review · Reviewer_72cG · 2025-03-12

**Overall Recommendation:** 3

**Summary:**

This paper proposes a loss function for image classification. It follows the idea of promoting better representation learning and proposes an improvement on mitigating the random consistency of existing methods. Properties such as unbiasedness and generalization bounds are theoretically investigated. Empirical evaluation of both accuracy and f-measures are conducted on multiple image datasets.

**Claims And Evidence:**

The first claim of the loss function proposal is well supported by the following contents.
The second claim of theoretical investigation is supported by section 4 and 5.
The last claim of a model proposal is also supported, but may narrow application scenes of the proposed method.

**Essential References Not Discussed:**

N/A

**Experimental Designs Or Analyses:**

Significance tests seems to evaluate the motivation of the proposed method, that is to mitigate the random consistency of existing methods. It would be nice to elaborate more on the connection between the significance tests and the motivation of the proposed method.

**Methods And Evaluation Criteria:**

The proposed methods aim to solve multi-class classification problems. The evaluation criteria of accuracy and f-measures are suitable, theoretical investgations also show indirect evidence for the proposed method.

**Other Comments Or Suggestions:**

N/A

**Other Strengths And Weaknesses:**

Strengths
- It is good to also discuss computation complexity of the proposed loss term.

Weakness
- This paper is not clear for some concepts and shows some difficulty for readers with less literature knowledge to follow.
- Section 6 seems to heavily restrict the selection of network architecture, hindering further applications.

**Questions For Authors:**

1. Existing methods seem to heavily rely on the design and selection of non-informative matrices. Is there any general summary on this point?
2. Please elaborate more on the motivation of the proposed loss function. How it "can address the shortcomings of dinfor(K) and dSED(K)" and how it can mitigate random consistency?
3. It says random consistency has been shown to exists in consistency measures. What is the detailed definition for random consistency, how is harms learning and how it exists in consistency measures?

**Relation To Broader Scientific Literature:**

This paper relates to the broad literature of classification and loss function design. It also draws attention on representation learning.

**Theoretical Claims:**

I checked the section 4 and 5 for theoretical claims.

---

> ### Author Rebuttal · Authors · 2025-03-31
>
> Dear Reviewer 72cG,
>
> We are very grateful for your valuable comments and questions. The responses are as follows:
>
> **Response to Claims And Evidence:** The proposed loss function serves as a universal quality measure for similarity matrices, which are foundational elements across various learning paradigms. Its generalized formulation enables applications in training deep network, metric learning, kernel learning and so on. To demonstrate the broad applicability, we applied it to train advanced network architectures (ViT and CNN). The experimental results and detailed settings are listed in our response to Reviewer WSCE Weakness 1.
>
> **Response to Experimental Designs Or Analyses:** By incorporating the random consistency mitigation mechanism (Eq. 5), we have enhanced the homogeneity and unbiasedness of the loss function, thereby improving the model's generalization capability. Significance tests demonstrate that this improvement maintains stable performance advantages across different data partitions, confirming that the enhancement is not attributable to random factors.
>
> **Response to Weaknesses 1:** This is an excellent suggestion. To enhance completeness and readability, we will add both a conceptual introduction to RC and an expanded literature review in the appendix of the next edition.  For further details, please see Question 3.
>
> **Response to Weaknesses 2:** This is a constructive suggestion. Indeed, the opening paragraph of Section 6 has some deficiencies in clarifying the motivation and scope behind our choice of network architecture. In fact, to validate the effectiveness of the loss function, we deliberately opted the proposed loss function for the most basic fully connected network structure. As mentioned earlier, our proposed loss function has relatively broad applicability. In the revised version, we will explicitly elaborate on the scope and adaptability of the method in section 6.
>
> **Response to Question 1:** The dependency of existing methods on uninformative matrices can be derived from Eq. 1, since  they measure the information content by computing the distance from uninformative matrices (e.g., identity matrix or full one matrix).
>
> **Response to Question 2:** The motivation of this paper is to propose a loss function ($d_{PSED}$) for measuring the information content of similarity matrices when ground-truth labels are available. The shortcoming of $d_{infor}(\mathbf{K})$ lies in its fundamental inability to properly assess the block-diagonal structure of similarity matrices, as evidenced by Property 1. To address this issue, $d_{PSED}$ introduces the ground-truth adjacency matrix as the reference standard, thereby enabling quantitative assessment of how closely the similarity matrix approximates the ideal block structure. The limitation of $d_{SED}(\mathbf{K})$ manifests as differential biases toward identity matrix $I$ versus all-ones matrix $J$ across varying class distributions. Through mitigating random consistency, we obtain $d_{PSED}$, a weighted element-wise dissimilarity metric for similarity matrices that achieves unbiased evaluation of both $I$ and $J$. The $d_{PSED}$ mitigates the random consistency issue by subtracting the expectation of $d_{SED}(\mathbf{K})$ under random label permutations (Eq.5).
>
> **Response to Question 3:** Consistency metrics measure the agreement between two random variables, while random consistency(RC) refers to spurious agreement arising purely from randomness[1]. A canonical manifestation of RC occurs when examinees achieve measurable test scores solely via random response patterns. The mechanisms by which RC harms the learning process include evaluation distortion, optimization misguidance, and generalization barriers.  Failure to deduct the RC baseline may lead to overestimating the model’s actual consistency performance (e.g., an original consistency score of 0.6 vs. a random baseline of 0.2 means the true effective consistency should be 0.4). When loss functions include RC without proper correction, they can induce optimization bias, causing algorithms to spuriously improve consistency metrics by overfitting to noise [2] or data bias[3,4] instead of learning genuine data patterns.  These would consequently impair the model's generalization ability.
>
> [1]Wang J, Qian Y, Li F, et al. Generalization performance of pure accuracy and its application in selective ensemble learning. IEEE Transactions on Pattern Analysis and Machine Intelligence, 2022, 45(2): 1798-1816.
>
> [2]Wang J, Qian Y, Li F. Learning with mitigating random consistency from the accuracy measure. Machine Learning, 2020, 109: 2247-2281.
>
> [3]Li J, Qian Y, Wang J, et al. PHSIC against random consistency and its application in causal inference. IJCAI, 2024, 5(10): 15-20.
>
> [4]Vinh N X, Epps J, Bailey J. Information theoretic measures for clusterings comparison: Variants, Properties, Normalization and Correction for Chance. Journal of Machine Learning Research, 2010, 11: 2837–2854.

---

### Decision · Program_Chairs · 2025-05-01

**Decision:**

Accept (poster)

**Comment:**

All three reviewers provided positive ratings after the rebuttal stage, with one reviewer upgrading their initial score. The initial concerns primarily centered on the completeness of ablation studies, implementation details, and the need for additional experimental results. During the post-rebuttal discussion phase, the reviewers expressed satisfaction with the authors’ responses and the revised manuscript. Upon careful examination of the paper, rebuttal, and reviewer discussions, the AC concurred with the reviewers’ assessment, acknowledging the soundness and effectiveness of the proposed method and subsequently recommending acceptance.

However, the manuscript still requires substantial revision—particularly in terms of writing quality and clarity and more comprehensive experimental comparisons.